# Geometry of Uncertainty: Learning Metric Spaces for Multimodal State Estimation in RL

**Alfredo Reichlin, Adriano Pacciarelli, Danica Kragic & Miguel Vasco**
Division of Robotics, Perception, and Learning
KTH Royal Institute of Technology
`{alfrei,adrianop,dani,miguelsv}@kth.se`

## Abstract

Estimating the state of an environment from high-dimensional, multimodal, and noisy observations is a fundamental challenge in reinforcement learning (RL). Traditional approaches rely on probabilistic models to account for the uncertainty, but often require explicit noise assumptions, in turn limiting generalization. In this work, we contribute a novel method to learn a structured latent representation, in which distances between states directly correlate with the minimum number of actions required to transition between them. The proposed metric space formulation provides a geometric interpretation of uncertainty without the need for explicit probabilistic modeling. To achieve this, we introduce a multimodal latent transition model and a sensor fusion mechanism based on inverse distance weighting, allowing for the adaptive integration of multiple sensor modalities without prior knowledge of noise distributions. We empirically validate the approach on a range of multimodal RL tasks, demonstrating improved robustness to sensor noise and superior state estimation compared to baseline methods. Our experiments show enhanced performance of an RL agent via the learned representation, eliminating the need of explicit noise augmentation. The presented results suggest that leveraging transition-aware metric spaces provides a principled and scalable solution for robust state estimation in sequential decision-making.

## 1 Introduction

Estimating the state of the environment from high-dimensional observations is a key challenge in Reinforcement Learning (RL). Compact low-dimensional representations of the sensory information have been shown to dramatically improve the performances of data-driven agents in synthetic (Anand et al., 2019) and real-world (Finn et al., 2016; Florensa et al., 2019) settings. In realistic scenarios, sensory information can be unreliable due to external noises or failures. In such cases, estimating the state of the system along with a measure of uncertainty allows for robust control, (Ackermann et al., 1993). When having access to multiple sensors, state estimation can become more precise as these can compensate for one another. This, however, requires a more complex design of a multimodal robust state estimation model.

Optimal solutions can be found by applying a Bayesian formulation to the problem. Bayesian filtering techniques allow for the maximum a-posteriori estimate of the state space even in case of uncertainties. These, however, require restrictions in terms of modeling the state distributions (Kalman, 1960) or using expensive Monte Carlo approaches (Gordon et al., 1993). Moreover, they generally require an observation model and prior knowledge of the functional form of the uncertainty (Thrun, 2002). In this regard, deep learning solutions have shown promising results on the deterministic representation of high-dimensional observations (Bengio et al., 2013) and complex dynamics (Hochreiter, 1997). Regarding uncertainty estimation, approaches included either training under noisy conditions by using variational methods (Krishnan et al., 2015) or using a reformulation of Gaussian processes (Garnelo et al., 2018). Reliable state estimation under uncertainty remains, however, an open challenge.

We address the problem of state representation from multi-sensory observations for RL. We propose to learn a representation that aligns the sensory modalities and correlates temporal distances with

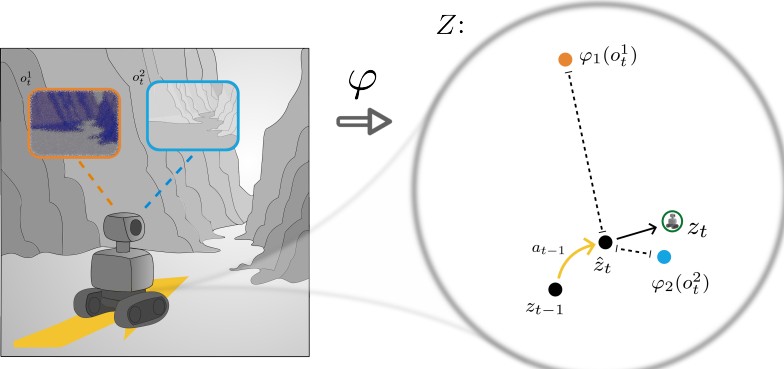

Figure 1: We propose METRICMM, a novel state estimation model from noisy multimodal observations. Each observation ($o_1, o_2$ on the left) is mapped into a joint metric space ($Z$ on the right) where distances from the latent dynamics prediction ($\hat{z}_t = \varphi_T(z_{t-1}, a_{t-1})$) are correlated with their uncertainty.

Euclidean distances. The key idea of this work is to recast the problem of uncertainty estimation geometrically in a metric space. This, in turn, allows to significantly simplify the problem of state estimation, Figure 1. The contributed model, Metric Learning for Multimodal State Estimation (METRICMM)[1], consists of an encoder for each modality and a latent transition model trained with simple contrastive learning and prediction losses. We empirically show that the proposed representation can be used to train an RL agent on a variety of tasks. Moreover, we demonstrate how the proposed state estimate is robust to sensor noise of arbitrary nature without being trained on any of these. Our work makes the following contributions:

1. **Metric-consistent multimodal state space.** We introduce METRICMM, which aligns sensory modalities in a shared latent space where temporal proximity corresponds to Euclidean distance, enabling simple geometric reasoning for control.

2. **Robustness under unseen corruptions.** We empirically demonstrate how METRICMM maintains high returns under seven corruption families (unseen during training), including settings where two of three modalities are corrupted, consistently outperforming fusion and representation baselines.

## 2 RELATED WORK

Representation learning describes the problem of extracting a meaningful representation from high-dimensional observations (LeCun et al., 2015). This is of particular interest in reinforcement learning, where the dimensionality of the observations scales exponentially with the data needed for learning (Kober et al., 2013). State representation learning, (Lesort et al., 2018), refers to those representations that describe the underlying state of a sequential decision-making problem. Early work in this direction has focused on estimating a latent representation that is low dimensional and invariant to distractors by using a reconstruction loss Munk et al. (2016); Mnih (2016) or contrastive methods Laskin et al. (2020). These approaches, however, lose the original structure of the state space and can't be generalized to other kinds of disturbances in the observation space.

In the case of reinforcement learning, data has a temporal structure due to the sequential nature of the problem. Model-based approaches make use of this structure to guide the learning of the representation and the policy. This can be done by learning a transition model concurrently with the representation to either facilitate the learning of a policy (Lillicrap, 2015; Haarnoja et al., 2018) or planning (Ha & Schmidhuber, 2018; Hafner et al., 2019b; Janner et al., 2021). Moreover, this temporal bias has been used explicitly to guide the structure of the latent space of the learned representation to simplify the control problem (Watter et al., 2015; Zhang et al., 2019; Eysenbach et al.,

---

[1]Code can be found at `https://github.com/reichlin/MetricMultiModal`

2022). While these methods use additional biases to simplify the learning problem or to recover additional properties, they do not offer a viable solution to the problem of noise robustness.

Bayesian filtering offers a solution to the problem of noise robustness in state estimation (Haykin, 2004). Classic methods of filtering offer provably optimal solutions in this direction but tend to be quite restrictive on the assumptions needed and the class of problems they can be applied to. Some of these assumptions have been successfully addressed by merging these models with modern versions of deep representation learning (Krishnan et al., 2015; Karl et al., 2016). These models, however, assume a noisy dataset to learn the uncertainty of the transition model and the observation model and restrict the model of the state estimate to be in a known form like a Gaussian distribution. Moreover, two major limitations are the need to learn a generative model for the update step and the inability to handle multimodal observations. In Haarnoja et al. (2016), they propose the use of a discriminative encoder for the observations to avoid learning the generative model. In Liu et al. (2023), they overcome the multimodality problem by substituting the Kalman Gain with a Transformer's attention module on the learned encoding of the different modalities. Contrary to these methods, our proposed model doesn't need to estimate the uncertainty explicitly allowing us to be agnostic to the kind of noise. Moreover, multimodality is easily addressed by aligning the representation with an invariant loss.

A generalization of these methods is described with the term State Space Models (SSM) where transitions and observations as well as noise are relaxed to arbitrary functions (Billings, 2013). Recurrent State Space Models (RSSM) learn explicitly the temporal relation of data using autoregressive models (Hafner et al., 2019a), or sequence-to-sequence models (Becker et al., 2024). Uncertainty can be taken into account using a probabilistic version of this (Doerr et al., 2018). This has been done using variational methods and imposing a known form of the probability estimate in the latent space using Variational AutoEncoders (VAE) (Kingma, 2013). Alternative solutions to avoid the reconstruction loss include the use of Prototypes (Deng et al., 2022) or contrastive methods (Becker et al., 2023). Similar to Bayesian filtering methods, uncertainty estimation remains a key challenge for these algorithms. Other forms of uncertainty estimation include Neural Processes (Garnelo et al., 2018; Jung et al., 2024), and Energy functions (Zhang et al., 2023). These have, however, been applied only to supervised learning settings.

Metric learning in the context of reinforcement learning has recently begun to attract growing interest. Steccanella & Jonsson (2022) introduce the concept of temporal distance in a POMDP setting and propose a novel objective to approximate these distances in a normed space. This, in turn, can be used for planning either via model predictive control or reward shaping. Similar normed spaces have been proposed to derive an exploration policy (Park et al., 2023), a foundation for general data-driven policies (Park et al., 2024), or learning an offline policy from sub-optimal demonstrations (Reichlin et al., 2026). Eysenbach et al. (2022) use a learned metric to estimate the discounted state occupancy measure of a policy within a policy-improvement loop. To address the problem of asymmetry in general POMDPs, Wang et al. (2023) extends this formulation to quasimetric representations and shows how these can be used to learn an optimal value function for goal-conditioned offline RL. Overall, these methods leverage metric (or quasimetric) structure primarily to facilitate policy learning or planning, whereas we instead exploit such spaces to address the problem of uncertainty estimation under sensory noise in multimodal state estimation.

## 3 BACKGROUND

Throughout the rest of the paper, we assume a Partially Observable Markov Decision Process (POMDP) defined by the tuple $(S, O^{1:N}, A, T, r, \gamma)$. Here, $S$ denotes the true underlying Markovian state space of the environment, $O_{1:N}$ denotes the $N$ available observation modalities, and $A$ the action space. The transition function $T : S \times A \to S$ is assumed to be deterministic. The objective is to maximize the cumulative reward given by the function $r : S \times A \to \mathbb{R}$, scaled by the discount factor $\gamma$. Each observation $o^i$ provides a potentially noisy measurement of the true state $s$. We assume that each observation is sampled from an unknown stochastic process, with independent noise affecting each modality, i.e. $o_t^i \sim p_i(o_t^i \mid s_t)$.

The overall goal of RL is to estimate a policy that maximizes the expected cumulative reward. When $s$ is not directly accessible, the policy is generally conditioned on the available observations, i.e., $O^{1:N}$. Representation learning for RL can be defined as finding a suitable latent

representation $z$ for the sensory observations, such that the policy, $\pi : Z \rightarrow A$, achieves the maximum possible expected cumulative reward. That is, preserving the information necessary for optimal decision-making while discarding task-irrelevant noise. Stochasticity in the observations requires modeling the representation as an estimation process which can be formulated as a Bayesian filtering problem. Via the Markov assumption, we can write the estimation process recursively through Bayes, i.e. $p(z_t \mid z_{t-1}, a_{t-1}, o_t^{1:N}) = p(o_t^{1:N} \mid z_t)p(z_t \mid z_{t-1}, a_{t-1})/p(o_t^{1:N})$. This is generally intractable when assumptions on the functional form of these probabilities cannot be made. Here, we simplify the estimation process and consider a deterministic representation of the state, as such we consider the maximum a-posteriori (MAP) estimate of the state, i.e. $z_t = \arg\max_{z_t} p(z_t \mid z_{t-1}, a_{t-1}, o_t^{1:N}) = \arg\max_{z_t} p(o_t^{1:N} \mid z_t)p(z_t \mid z_{t-1}, a_{t-1})$. In literature, the second term ($p(z_t \mid z_{t-1}, a_{t-1})$) is generally referred to as the prediction step while the first one ($p(o_t^{1:N} \mid z_t)$) as the update.

## 4 METHOD

We propose a novel approach to learn a latent representation of multi-sensory observations, along with a latent transition model, that enables robust state estimation independently of the nature of observation noise. Our method learns a metric space in which distances between latent states correlate with the minimum number of actions needed to transition between them in the environment. This provides a geometric interpretation of uncertainty, simplifying state estimation.

### 4.1 LATENT STATE REPRESENTATION

In an ideal scenario, having access to a perfect representation of the environment's dynamics would be sufficient to track the current state. However, in practice, approximation errors in the learned transition model and potential stochasticity in the POMDP dynamics must be accounted for. In sequential decision-making problems, these errors can compound over time, leading to divergence in the state estimate (Ross et al., 2011). To mitigate this issue, we incorporate information from multiple sensor modalities to refine the state estimate at each step, akin to the Bayesian filtering formulation. However, traditional filtering approaches require explicit uncertainty modeling, which may not generalize well across varying noise conditions.

Instead, we propose to structure the latent space such that state transitions and sensory observations can be integrated without explicit probabilistic modeling. The key idea behind this work is that the uncertainty induced by transition model errors is generally *local* to its predictions, not in the raw observation space but in a space where distances are induced by the system's dynamics. Specifically, we argue that the transition model's uncertainty should be interpreted in a space where distances correspond to the minimum number of actions required to transition between states. This motivates the construction of a metric space that aligns with the dynamics of the environment.

To formalize this, we define a metric space $\mathcal{M} = (Z, \|\cdot\|_2)$ induced by an ideal injective map $\varphi : S \rightarrow Z$, where $Z \subseteq \mathbb{R}^m$ is a vector space, and distances are given by the Euclidean norm. Following previous work (Steccanella & Jonsson, 2022; Park et al., 2024; Eysenbach et al., 2022; Wang et al., 2023; Park et al., 2023), we define $Z$ such that distances in this space are correlated with the minimum number of actions needed to transition between their corresponding environment states, i.e., temporal distances. Inducing a norm as the measure of distance does not allow for asymmetry, i.e., a quasimetric. As already noted by Steccanella & Jonsson (2022), such a formulation can only capture a symmetrized approximation of these distances, e.g., $\min\{d(s_1, s_2), d(s_2, s_1)\}$. Injectivity between the two spaces allows us to use the formalism of MDP Homomorphisms (Ravindran & Barto, 2001; Van der Pol et al., 2020). In the case of POMDPs, we do not have direct access to the state of the system and thus have to rely on sensory observations and dynamic predictions. As such, we can define an approximation of the dynamics of the environment in this new latent space, i.e. $\varphi_T : Z \times A \rightarrow Z$. Given this structure, we make the following assumption:

**Transition Error Assumption**: The transition error at time $t$ is constrained such that the predicted latent state $\varphi_T(z_{t-1}, a_{t-1})$ lies within a small ball of states that are close in terms of action-based distance. That is, there exists an $\epsilon$-radius region in the latent space such that:

$$z_t \in \mathcal{B}(\varphi_T(z_{t-1}, a_{t-1}), \epsilon), \tag{1}$$

where $\mathcal{B}(z, \epsilon) = \{z' \in Z \mid d(z, z') \leq \epsilon\}$.

Intuitively, while the transition model may introduce small prediction errors, these errors generally remain within a neighborhood of states that require similar sequences of actions to transition between. Nevertheless, the learned transition model accumulates errors over time. This requires the integration of sensory information to correct the latent state estimate. Each modality $O^i$ provides an independent estimate of the state. In this regard, we define a deterministic mapping from each observation space to the above-defined metric space $Z$:

$$\varphi_i : O^i \rightarrow Z, \quad \forall i \in [1, N] \tag{2}$$

where $N$ represents the number of sensor modalities. We seek to learn these mappings such that the latent representation is aligned between them and respects the notion of metric space previously defined.

## 4.2 SENSOR FUSION VIA INVERSE DISTANCE WEIGHTING

The information from the sensors can be used to correct the prediction of the latent transition model on the state of the environment. However, due to noise, different modalities might be more or less reliable at any given time. We can make use of the geometry induced by the metric space to approximately model this uncertainty. Since we do not assume prior knowledge of noise distributions, we propose an adaptive weighting scheme based on the notion of distance induced by the metric space. The key intuitions are:

- If an observation's latent encoding $\varphi_i(o^i_t)$ is *close* to the latent transition prediction $\varphi_T(z_{t-1}, a_{t-1})$, then it is more likely to be an accurate state estimate.
- Conversely, if an observation encoding is *far* from the prediction, it is likely corrupted by noise or uninformative.

Thus, we weigh each modality's contribution using the inverse of its latent space distance to the transition model estimate. The final estimated state is:

$$z_t = \left( \sum_i \frac{1}{\|z^i_t - \hat{z}_t\|_2 + \delta} \right)^{-1} \sum_i \frac{z^i_t}{\|z^i_t - \hat{z}_t\|_2 + \delta}, \tag{3}$$

where $\hat{z}_t = \varphi_T(z_{t-1}, a_{t-1})$, $z^i_t = \varphi_i(o^i_t)$ and $\delta = 10^{-5}$. This formulation follows a MAP principle, where more confident estimates (i.e., those that agree with the transition model) receive higher weights. This enables robustness without requiring explicit noise modeling.

## 4.3 LEARNING THE LATENT REPRESENTATION

We introduce three loss functions to enforce the desired metric structure of the latent space. We do not require noisy observations during training as we do not need an explicit estimate of the uncertainty. Whenever we refer to $\bar{z}_t$ or $\bar{z}_{t+1}$ in the loss functions, we mean the average of the sensor encodings:

$$\bar{z}_t = \frac{1}{N} \sum_{i=1}^{N} \varphi_i(o^i_t), \tag{4}$$

Assuming no noise in the sensory observations, the mean is equivalent to the state estimate in Equation 3.

**Contrastive Temporal Distance Loss** We enforce temporal consistency by structuring the latent space such that successive states remain close, while randomly sampled states are further apart:

$$\mathcal{L}_+ = \mathbb{E}[(\|\bar{z}_{t+1} - \bar{z}_t\|_2 - 1)^2] \tag{5}$$

$$\mathcal{L}_- = \mathbb{E}[-\log(\|\bar{z}_r - \bar{z}_t\|_2)] \tag{6}$$

where $z_r$ is a randomly sampled state. A similar formulation for different applications was proposed in Wang et al. (2023); Park et al. (2023; 2024). The term $\mathcal{L}_-$ ensures that the latent representation does not collapse into a trivial solution where all states are mapped to the same point. By encouraging larger distances between random state pairs, we preserve meaningful geometry in the representation space. This term, however, is bounded by the triangular inequality given the positive term of the loss, i.e. $\mathcal{L}_+$. As pointed out in Wang et al. (2023), maximizing negative distances, with $\mathcal{L}_+$ as a local constraint, effectively spreads out every state as much as possible. In turn, this results in a representation where temporal distances are recovered also for non-adjacent pairs.

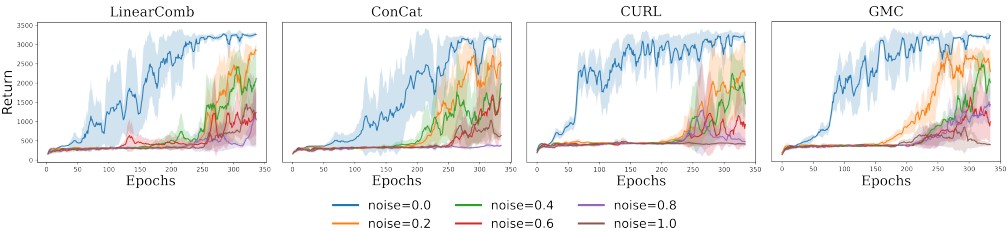

Figure 2: Mean and standard deviation over 5 seeds of the training return for a SAC agent on the Hopper-v5 environment. The policies are trained with different state estimation modules (LinearComb, ConCat, CURL, GMC) and different amounts of Gaussian noise on the observations. With an increase in noise, the expected average return sensibly decreases for all the estimators. Epochs are in thousands.

**Latent Transition Loss**   To ensure consistency between the learned transition model and observed transitions, we minimize:

$$\mathcal{L}_T = \mathbb{E}[(\varphi_T(\bar{z}_t, a_t) - \bar{z}_{t+1})^2] \tag{7}$$

This enforces that the transition model accurately captures environment dynamics.

**Multimodal Invariance Loss**   To align representations across sensor modalities, we introduce an invariance loss:

$$\mathcal{L}_{inv} = \mathbb{E}[(\varphi_i(o_t^i) - \varphi_j(o_t^j))^2] \quad \forall i, j \in [1, N] \tag{8}$$

This ensures that each modality is mapped to the same learned latent metric space.

## 4.4   Integration with Reinforcement Learning

The representation objectives described above are combined linearly to form the overall loss:

$$\mathcal{L} = \mathcal{L}_T + \lambda_1 \mathcal{L}_+ + \lambda_2 \mathcal{L}_- + \lambda_3 \mathcal{L}_{inv}, \tag{9}$$

where $\lambda_1, \lambda_2, \lambda_3$ are scalar weighting coefficients. The learned latent representation $\mathbf{z}$ serves as the input to any standard reinforcement learning algorithm and is conceptually independent of the specific policy optimization method used. In practice, we jointly optimize the representation loss $\mathcal{L}$ and the reinforcement learning objective in an end-to-end fashion. Unlike traditional approaches that rely on explicit noise augmentation or uncertainty modeling, our formulation inherently accounts for observation uncertainty through the geometry of the latent space, simplifying training while improving robustness to corrupted or missing sensory inputs.

## 5   Experiments

**Scope and setup.**   We study whether multimodal representations can sustain control performance under severe observation perturbations and cross–modal mismatch. Our evaluation spans two suites. (i) `MuJoCo`: Hopper-v5, HalfCheetah-v5, Ant-v5, Walker2d-v5, Humanoid-v5, and InvertedPendulum-v5 with synchronized RGB and depth streams. (ii) `Fetch`: 7-DoF manipulation (FetchPickAndPlace-v4, FetchSlide-v4) with RGB, depth, and point clouds. Unless otherwise stated, we train Soft Actor–Critic (SAC) end-to-end on top of the representation module and report mean return $\pm$ standard deviation over 5 seeds. All architectural and optimization details are deferred to the Appendix.

**Corruptions and evaluation protocol.**   To probe robustness, we inject seven families of perturbations at *test* time (unseen during training): *Gaussian* (Hendrycks & Dietterich, 2019), *Salt-and-Pepper* (Hendrycks & Dietterich, 2019), *Patches* (Becker et al., 2023; Grigsby & Qi, 2020; Hansen & Wang, 2021), *Puzzle* (Bucci et al., 2021; Noroozi & Favaro, 2016), *Texture* (Becker et al., 2023; Liu et al., 2023; Hansen & Wang, 2021), *Failure* (Poklukar et al., 2022; Skand et al., 2024), and *Hallucination* (Zhang et al., 2020). These are applied with different probabilities to study how fast performances deteriorate. For `MuJoCo`, we corrupt one modality at a time to isolate each sensor modality contribution to the overall state estimate; for `Fetch`, we additionally consider settings

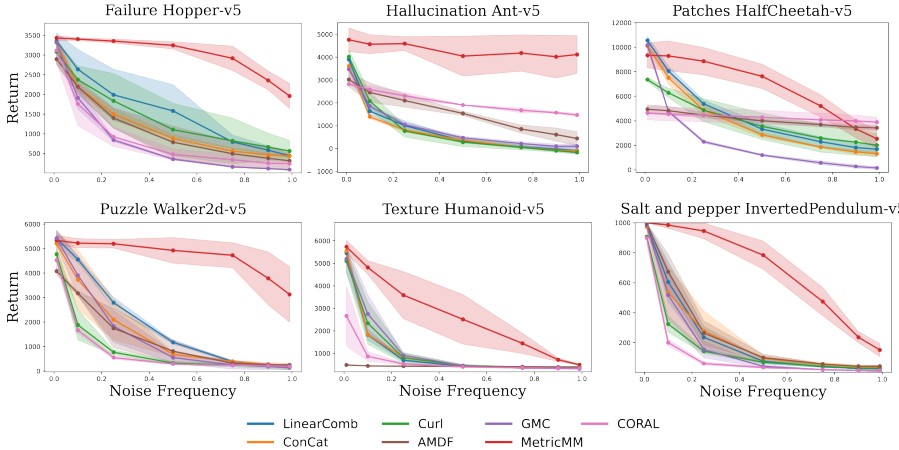

Figure 3: Mean and standard deviation over 5 seeds and 50 trajectories of the testing return for a SAC agent on the Mujoco suite. The policies are tested with different state estimation modules and different amounts of noise (perturbations of one modality at a time). METRICMMis the only estimator that allows for a consistent return with high-frequency perturbations.

Table 1: Return of multimodal fusion methods under Patch corruptions applied simultaneously to two modalities on `Fetch–PickAndPlace`, for increasing corruption probabilities. METRICMM preserves strong control performance across all corruption levels, while alternative fusion strategies degrade much more rapidly.

| Model | 0.1 | 0.25 | 0.5 | 0.75 | 0.9 | 0.99 |
|---|---|---|---|---|---|---|
| LinearComb | -0.89 ± 0.02 | -1.99 ± 1.14 | -1.76 ± 1.35 | -2.57 ± 2.11 | -2.67 ± 1.56 | -2.17 ± 1.28 |
| Concat | -0.04 ± 1.63 | -1.13 ± 0.9 | -2.84 ± 0.76 | -2.06 ± 0.43 | -2.53 ± 0.63 | -2.43 ± 0.24 |
| CURL | 1.43 ± 0.32 | -1.55 ± 0.93 | -3.76 ± 1.94 | -3.51 ± 0.59 | -3.4 ± 0.46 | -2.58 ± 0.38 |
| GMC | -0.01 ± 1.16 | -0.91 ± 0.27 | -2.04 ± 0.38 | -1.95 ± 0.67 | -1.88 ± 0.89 | -2.41 ± 1.24 |
| AMDF | **2.2 ± 1.32** | 0.93 ± 0.62 | -1.04 ± 0.54 | -1.75 ± 0.59 | -2.16 ± 0.35 | -2.34 ± 0.31 |
| CORAL | -0.36 ± 0.43 | -0.86 ± 0.23 | -1.51 ± 0.99 | -1.23 ± 0.64 | -1.38 ± 0.86 | **-1.47 ± 0.94** |
| MetricMM | 1.91 ± 1.09 | **1.87 ± 0.93** | **1.43 ± 1.14** | **0.92 ± 0.79** | **-0.91 ± 0.26** | **-1.47 ± 0.1** |

where two of the three modalities are simultaneously corrupted. For each (task, corruption) tuple, we evaluate 50 episodes per seed and aggregate across seeds to obtain performance–severity curves (Fig. 3).

**Baselines.** We compare against six representative fusion/representation methods, matching encoder capacity and tuning budget:

- **Linear Combination** (LinearComb): a learned linear map that combines per-modality latent features.
- **Concatenation** (ConCat): feature concatenation followed by a shared projection.
- **CURL** (Laskin et al., 2020): contrastive learning on augmented views at each timestep, without explicit cross-modal fusion.
- **GMC** (Poklukar et al., 2022): cross-modal contrastive alignment to learn a shared embedding.
- **$\alpha$-MDF** (Liu et al., 2023): an attention-weighted differentiable Bayesian filter that fuses modality-specific encoders into a latent state.
- **CORAL** (Becker et al., 2023): joint latent space learned via per-modality reconstruction and temporal contrastive alignment.

All models are trained jointly with SAC under identical data budgets, replay settings, and evaluation schedules. Additional implementation details, architectures and hyperparameters are deferred to the Appendix.

Table 2: Return of multimodal fusion methods under Failure corruptions applied simultaneously to two modalities on `Fetch`–Slide, for increasing corruption probabilities. METRICMM preserves strong control performance across all corruption levels, while alternative fusion strategies degrade much more rapidly.

| Model | 0.1 | 0.25 | 0.5 | 0.75 | 0.9 | 0.99 |
|---|---|---|---|---|---|---|
| LinearComb | $5.64 \pm 0.92$ | $3.31 \pm 0.98$ | $1.41 \pm 2.14$ | $-1.21 \pm 1.74$ | $-3.58 \pm 1.17$ | $-3.36 \pm 0.35$ |
| ConCat | $7.01 \pm 1.64$ | $5.5 \pm 1.4$ | $4.01 \pm 0.48$ | $0.06 \pm 1.54$ | $-1.61 \pm 1.73$ | $-1.49 \pm 1.2$ |
| CURL | $7.43 \pm 0.43$ | $5.34 \pm 0.83$ | $1.89 \pm 2.35$ | $-1.36 \pm 3.33$ | $-2.24 \pm 1.87$ | $-3.69 \pm 3.21$ |
| GMC | $5.17 \pm 0.4$ | $1.24 \pm 3.33$ | $0.12 \pm 1.78$ | $-1.77 \pm 0.63$ | $-2.72 \pm 1.02$ | $-3.11 \pm 1.59$ |
| AMDF | $3.05 \pm 1.62$ | $2.76 \pm 1.61$ | $-0.48 \pm 1.0$ | $-0.67 \pm 1.93$ | $-1.98 \pm 1.65$ | $-2.41 \pm 2.25$ |
| CORAL | $4.7 \pm 2.92$ | $2.28 \pm 0.63$ | $-1.16 \pm 0.71$ | $-1.39 \pm 0.8$ | $-1.87 \pm 1.8$ | $-1.79 \pm 0.97$ |
| MetricMM | $\mathbf{9.26 \pm 0.21}$ | $\mathbf{7.77 \pm 2.02}$ | $\mathbf{5.95 \pm 2.36}$ | $\mathbf{4.55 \pm 2.96}$ | $\mathbf{1.33 \pm 2.49}$ | $\mathbf{-0.23 \pm 3.58}$ |

**Results on `MuJoCo`: robustness to single-modality corruption.** Figure 3 summarizes performance as the perturbation probability increases for six tasks and diverse corruptions. We observe three consistent trends. **(i) Robustness slope.** Our method (METRICMM) exhibits the flattest degradation, preserving a large fraction of the clean performance up to mid/high severities across tasks; in contrast, simple fusion baselines (LinearComb/Concat) degrade steeply even for low frequency of corruptions, indicating that naive aggregation does not resolve cross-modal disagreements. **(ii) Method ordering is stable across corruptions.** Across failure, hallucination, texture, puzzle, patches, and salt-and-pepper, METRICMM maintains the top curve while CURL and GMC are competitive at low frequencies but drop sharply once the corrupted modality dominates the fused signal. **(iii) High-DoF tasks are particularly sensitive.** On Humanoid-v5 and HalfCheetah-v5, the gap between METRICMM and the best baseline widens with an increased perturbation's frequency, suggesting that principled cross-modal consistency becomes increasingly important as control complexity grows.

**Results on `Fetch`: robustness under multi-modality corruption.** We next corrupt two of the three modalities on manipulation tasks. Tables 1 and 2 report returns for *patches* and *failure* corruptions, respectively, as the probability of corruption per time step increases. Two effects stand out. **(i) Majority corruptions.** Even when the majority of sensory channels are degraded, METRICMM retains substantially higher returns, e.g., on Fetch Slide with failure noise, it remains above zero reward until very high severities, while all baselines collapse much earlier (Table 2). **(ii) Graceful decay vs. collapse.** On Fetch Pick-and-Place with patches, baseline returns turn negative quickly as the probability increases, while METRICMM degrades gradually and remains competitive at intermediate frequencies (Table 1). These results indicate that METRICMM can prioritize and re-weight the remaining reliable modality when others fail.

**Training with noisy observations is not required (and can be harmful).** Figure 2 shows learning curves on Hopper-v5 when injecting noise during *training* for four of the baselines. While moderate noise can induce invariances, it consistently slows exploration and lowers asymptotic returns across preprocessors; severe noise significantly delays the onset of learning. Practically, this requires knowing the corruption family a priori and increases computation, whereas METRICMM attains robustness without any noisy training.

**Every modality provides useful information.** Not all observations are equally easy to exploit, and without explicit cross-modal alignment, a policy often latches onto the cheapest signal. In `Fetch`, the point-cloud stream is the most informative for precise geometry but also the most demanding to encode, making it especially vulnerable to perturbations. To quantify reliance, Table 3 reports performance when we corrupt *only one* modality at a high probability (0.99). When point clouds are the sole corrupted stream, most baselines exhibit little to no degradation, revealing a systematic over-reliance on RGB/depth and under-utilization of 3D structure. In contrast, our aligned representation distributes credit across modalities and shows a more uniform sensitivity profile, indicating that each sensor contributes meaningfully to the learned state.

**Temporally dependent noise.** In realistic scenarios, noise is not necessarily time independent. For example, a faulty sensor might produce persistent noisy observations for multiple consecutive time steps. In Figure 4 we present results on the `Fetch`suite for persistent sensor failure perturbations under 3 and 10 consecutive frames. METRICMM remains the most reliable method.

Table 3: Return of multimodal fusion methods under Failure corruptions applied to a single modality with probability 1 on `Fetch`–Slide. METRICMM remains robust independently of the dropped modality.

| Model | all modalities | image only | depth only | point cloud only |
|---|---|---|---|---|
| Linear Comb | $-1.14 \pm 2.41$ | $-2.94 \pm 1.26$ | $-1.96 \pm 2.14$ | $9.27 \pm 0.17$ |
| ConCat | $1.53 \pm 2.91$ | $0.28 \pm 6.19$ | $-2.38 \pm 0.67$ | $7.37 \pm 2.11$ |
| CURL | $0.16 \pm 3.09$ | $-2.17 \pm 1.51$ | $-0.06 \pm 8.26$ | $8.15 \pm 0.94$ |
| GMC | $-1.76 \pm 1.31$ | $-1.13 \pm 1.10$ | $-4.52 \pm 4.01$ | $7.67 \pm 0.53$ |
| AMDF | $0.1 \pm 1.53$ | $-2.62 \pm 2.87$ | $-1.63 \pm 2.91$ | $3.80 \pm 2.28$ |
| CORAL | $-1.32 \pm 0.67$ | $-3.87 \pm 4.63$ | $-2.53 \pm 1.15$ | $7.56 \pm 2.03$ |
| MetricMM | $8.46 \pm 0.92$ | $7.29 \pm 1.39$ | $8.90 \pm 0.96$ | $7.87 \pm 0.65$ |

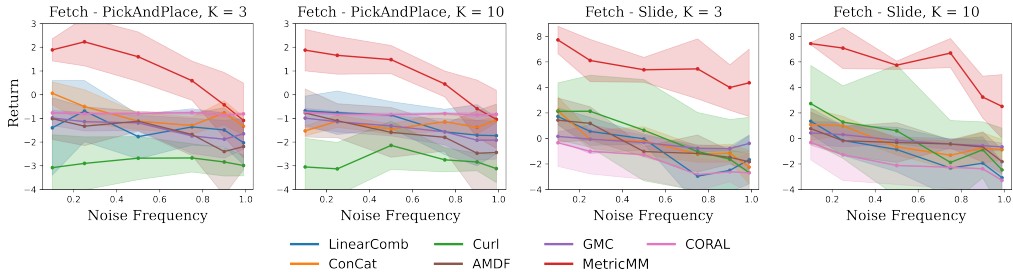

Figure 4: Mean and standard deviation over 5 seeds and 50 trajectories of the testing return for a SAC agent on the Fetch suite under time-persistent sensor failure. The policies are tested with different state estimation modules and different amounts of noise. Each time the noise is applied, it persists for either 3 or 10 consecutive frames ($K$) on that specific sensor modality. METRICMMis the only estimator that allows for a consistent return with high-frequency perturbations.

## 5.1 ABLATIONS

We study the effects of the different components of the representation loss. For this, we consider an additional one-dimensional pendulum environment where the goal is to swing the pendulum upside down, following (Silva et al., 2020). As sensor modalities, we rely on images of the scene as well as sounds generated by the pendulum swinging and captured by three evenly distributed receivers. The amplitude and frequency of the received signal give us information on both the position and velocity of the pendulum, thanks to the Doppler effect. For this experiment, we consider the performances of METRICMM against two versions of itself, one trained without the invariance loss term ($\mathcal{L}_{inv}$) and one without the metric loss pair ($L_+$ and $L_-$). With no invariance loss, the representation has no incentive to align the two modalities. Without the metric terms, on the other hand, the representation loses the temporal structure and gets arranged to minimize the transition loss which compresses the space considerably.

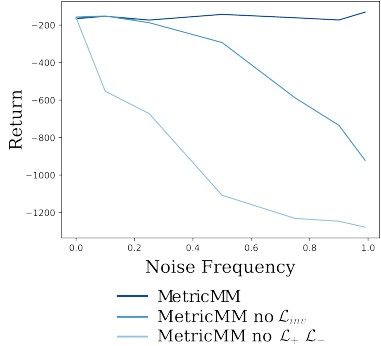

Figure 5: Average return for a SAC agent on the one-dimensional pendulum environment and increasing level of Gaussian noise. A METRICMM exhibits robust performances, both of the ablation variants degrade with an increase in noise.

In both cases, SAC is still able to learn a meaningful policy on top of these representations. However, the noise robustness ability is lost as depicted in Figure 5.

## 5.2 LIMITATIONS

Across locomotion and manipulation, METRICMM maintains strong control performance under severe and diverse corruptions, including cases where two of three modalities are degraded. The method remains robust without any noise-aware training, offering a practical route to resilient multimodal RL. Nevertheless, it has a few important limitations.

**Symmetric metric assumption.** In its current form, METRICMM defines distance in latent space via a norm, and is therefore restricted to *symmetric* spaces. This prevents it from exactly representing quasimetric structures that frequently arise in RL, where the *distance* from $s$ to $s'$ can differ from the distance from $s'$ to $s$ due to irreversibility or constraints. As a result, the learned metric can only provide a symmetric approximation of the true minimal-action distance. While this limits the theoretical fidelity of the representation, it is not catastrophic in practice. Distances, in this case, can still provide an approximation of uncertainty, e.g., pushing objects in the Fetch environments is directionally constrained. The policy, on the other hand, is not affected by this. The network does not explicitly rely on the measure of distance to estimate the optimal action. Nevertheless, generalizing the representation to a quasimetric would allow for a tighter uncertainty estimation.

**Deterministic dynamics.** As discussed in the background section, our formulation assumes deterministic dynamics in the environment. In the case of stochastic transitions, the representation considers every possible

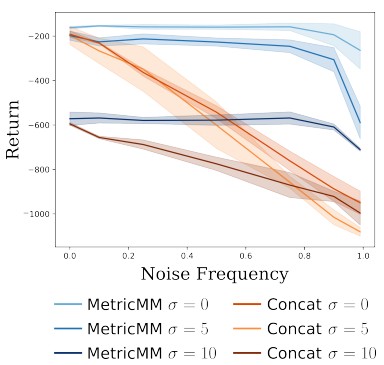

Figure 6: Average return for a SAC agent on the stochastic pendulum with Hallucination noise. With increased stochasticity ($\sigma$) the performances reduce, METRICMM remains robust while a ConCat baseline quickly degrades.

next state to be adjacent to the current one and thus embeds it at a unit distance in $Z$. Moreover, the latent transition model estimates the barycenter of this next-state distribution. In moderately stochastic conditions, this is possible in practice. Figure 6 illustrates the degradation in performance under Hallucination noise for a stochastic variation of the pendulum experiment for METRICMM compared to a ConCat baseline. While overall performance degrades as stochasticity increases, METRICMM retains a clear robustness advantage. Stochasticity in the dynamics does require an increase in the dimensionality of the learned space to accommodate for an increase in the number of adjacent next-state and might be unfeasible in practice. Moreover, modeling the latent transitions with a deterministic module results in a decrease in expressivity. Extending the framework to a general stochastic setting might require a different notion of distance (e.g., replacing the norm-based distance with a divergence between predictive distributions) and a different transition model.

## 6 CONCLUSION

We introduced a novel approach for robust state estimation in reinforcement learning by learning a structured latent space where distances reflect the minimum number of actions required to transition between states. This metric space formulation enables a geometric interpretation of uncertainty, eliminating the need for explicit probabilistic noise modeling. Our method integrates multi-sensory observations via inverse distance weighting, ensuring adaptive sensor fusion without prior knowledge of noise distributions. Additionally, contrastive and transition-consistency losses enforce temporal structure, while an invariance loss aligns representations across modalities. We empirically demonstrated the effectiveness of our approach on a diverse set of tasks, showing that it improves state estimation robustness in the presence of sensor noise and significantly enhances RL agent performance. Results confirm that the learned representation generalizes across different environments without requiring explicit noise augmentation. By leveraging the environment's dynamics within the latent space, our approach provides a scalable and principled solution for robust state estimation. Future directions include adapting the method to non-stationary environments, evaluating adversarial robustness, and extending it to real-world robotics.

## 7 ACKNOWLEDGEMENT

The work was enabled by the Berzelius resource provided by the Knut and Alice Wallenberg Foundation at the National Supercomputer Centre. We further thanks the Swedish Research Council and the Knut and Alice Wallenberg Foundation. We thank Jens Lundell for the simulation of the point clouds code.

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

## A APPENDIX

### A.1 EXPERIMENTAL DETAILS

Here, we provide the necessary details on the conducted experiments. For the `MuJoCo` suite, we augmented the observations to include both RGB images and depth images. For both, we used a resolution of 84×84. To ensure Markovianity in the observations, we stack 3 consecutive frames together for both the RGB images and the depth images. We do not use proprioceptive information. We keep the reward function to be the default one of the different environments. For the `Fetch` suite, we augmented the observations to include RGB images, depth images and point clouds. Both RGB and depth images have a resolution of 128×128. We generate the point cloud observations by projecting rays from the camera position, given the depth of the environment. This results in a variable number of points, each consisting of a 3D position and an RGB value. Again, for all the modalities, we stack 3 consecutive frames together. For these environments, we changed the reward function to be a linear combination of the negative Euclidean distance between the end-effector of the arm and the object and the negative Euclidean distance between the object and the goal (the latter scaled by the initial distance and multiplied by a factor of 10). For the Pendulum experiment, the dynamics are defined as:

$$\ddot{\theta}_t = \frac{3g}{2\ell} \sin \theta_t \; + \; \frac{3}{m\ell^2} \, a_t, \tag{10}$$

$$\xi_t \sim \mathcal{N}\left(0, \sigma^2\right), \tag{11}$$

$$\dot{\theta}_{t+1} = \left(\dot{\theta}_t + (\ddot{\theta}_t + \xi_t)\,\Delta t, \; \right), \tag{12}$$

$$\theta_{t+1} = \theta_t + \dot{\theta}_{t+1}\,\Delta t, \tag{13}$$

where $g = 10, \ell = 1, m = 1, \Delta t = 0.05$. The sound observations are generated by computing the frequency and amplitude of a sound originating from the tip of the swinging pendulum at a constant frequency. The sound is perceived by 3 receivers at each time step in the form of a frequency (modeled via the Doppler effect) and an amplitude (modeled via the inverse-square law).

For each model and each environment, we learn a Soft Actor-Critic agent to maximize the reward of the environment. The list of relevant hyperparameters is described in Table 4, these are the same for every experiment.

Table 4: SAC hyperparameters.

| Hyperparameter | Value |
| --- | --- |
| $\gamma$ | 0.99 |
| $\tau$ | 0.005 |
| Learning rate actor | 0.0003 |
| Learning rate critic | 0.001 |
| Replay buffer size | 100000 |
| Batch size | 256 |
| Initial $\alpha$ | 0.1 |
| Number of critics | 2 |
| Number of simulations between updates | 1 |
| Number of workers | 2 |
| Target entropy | - dim of actions |
| Number of hidden layers critic | 3 |
| Number of neurons per layer critic | 256 |
| Non-linear activation critic | ReLU |
| Number of hidden layers actor | 3 |
| Number of neurons per layer actor | 256 |
| Non-linear activation actor | ReLU |

For every experiment, we train the SAC agent together with its representation module end-to-end by minimizing the linear combination of all its losses. We train the agent until convergence, around 200 thousand steps for the `MuJoCo` environments and 400 thousand for the `Fetch` environments. After

convergence, we test the best-performing models for 50 trajectories. Noise is applied at random to any modality at every step (except the initial observation) with a different probability.

For every state estimation method we use a Convolutional Neural Network (CNN) architecture for the RGB and the depth images and a SetTransformer for the point cloud. The CNN consists of 3 convolutional layers with 32 filters each and a kernel size and stride respectively of [4, 2], [4, 2], [3, 1] with a fully connected output layer. Each layer is followed by a ReLU non-linear activation function. We encode each point cloud frame with a compact SetTransformer: 6-D points (XYZ+RGB) are linearly projected to width $d_{model}=64$, processed by two self-attention blocks (4 heads, Layer-Norm+residuals, feed-forward width $2\times$), and pooled via multihead attention with a single learned seed ($k=1$). Over a sequence of $T=3$ frames, the pooled 64-D summaries are concatenated and mapped with a linear layer to a latent $z$ of size $d_z=64$, then refined by a small MLP ($64\rightarrow64\rightarrow d_z$) to produce the final embedding.

Below is the list of hyperparameters used for METRICMM and the baselines:

- METRICMM: latent size $d_z = 64$. Per-modality encoders (CNN for images, Set Transformer for point clouds). The loss hyperparameters are all 1, i.e., $\lambda_T = \lambda_1 = \lambda_2 = \lambda_{inv} = 1$.

- LinearComb: per-modality encoders (CNN/Set Transformer) projecting to $d_z$, learned linear fusion over modality latents, capacity matched to ours (same $d_z$, same encoder widths).

- ConCat: same per-modality encoders to $d_z$; fusion by feature concatenation (final latent dimension scales with number of modalities).

- CURL: per-modality encoders to $d_z$ with a contrastive head, InfoNCE temperature $\tau = 0.2$, momentum target encoder with EMA $m=0.99$; standard cropping as image augmentations and gaussian noise as point cloud augmentations.

- GMC: per-modality encoders to a shared embedding with cross-modal contrastive alignment; InfoNCE temperature $\tau=0.3$; same $d_z$ and encoder widths as ours for fairness.

- AMDF: attention-weighted differentiable filter over modality encoders with recurrent latent state; latent size $d_z$ and filter MLP widths matched to ours; attention temperature/weights from the implementation defaults.

- CORAL: per-modality encoders with joint latent space trained via reconstruction + contrastive objectives; InfoNCE temperature $\tau=0.1$; same $d_z$ and encoder capacity as ours.

## A.2 NOISES

For all the environments we experiment with the following families of corruptions as noise. Figure 7 shows an example of the different noises for the `MuJoCo`Ant-v5 environment.

- Gaussian: linear combination with samples from a Gaussian distribution. For RGB and depth images: $\mathcal{N}(0, 25)$, for point clouds: $\mathcal{N}(0, 0.03)$. Can be applied to all the modalities.

- Salt-and-pepper: with probability 0.3 for `MuJoCo`and 0.8, each dimension of the observation is transformed to either the minimum or maximum possible value. Can be applied to all the modalities.

- Patches: a portion at random of the image is masked, 30% for `MuJoCo`observations and 50% for `Fetch`. Can be applied to RGB and depth images only.

- Puzzle: the image is divided into a 3 by 3 grid and reshuffled. Can be applied to RGB and depth images only.

- Texture: the background of the image is segmented and replaced with another image. Can be applied to RGB images only.

- Failure: the entire observation is set to 0. Can be applied to all modalities.

- Hallucination: the entire observation is substituted with another in-distribution observation from another trajectory. Can be applied to all modalities.

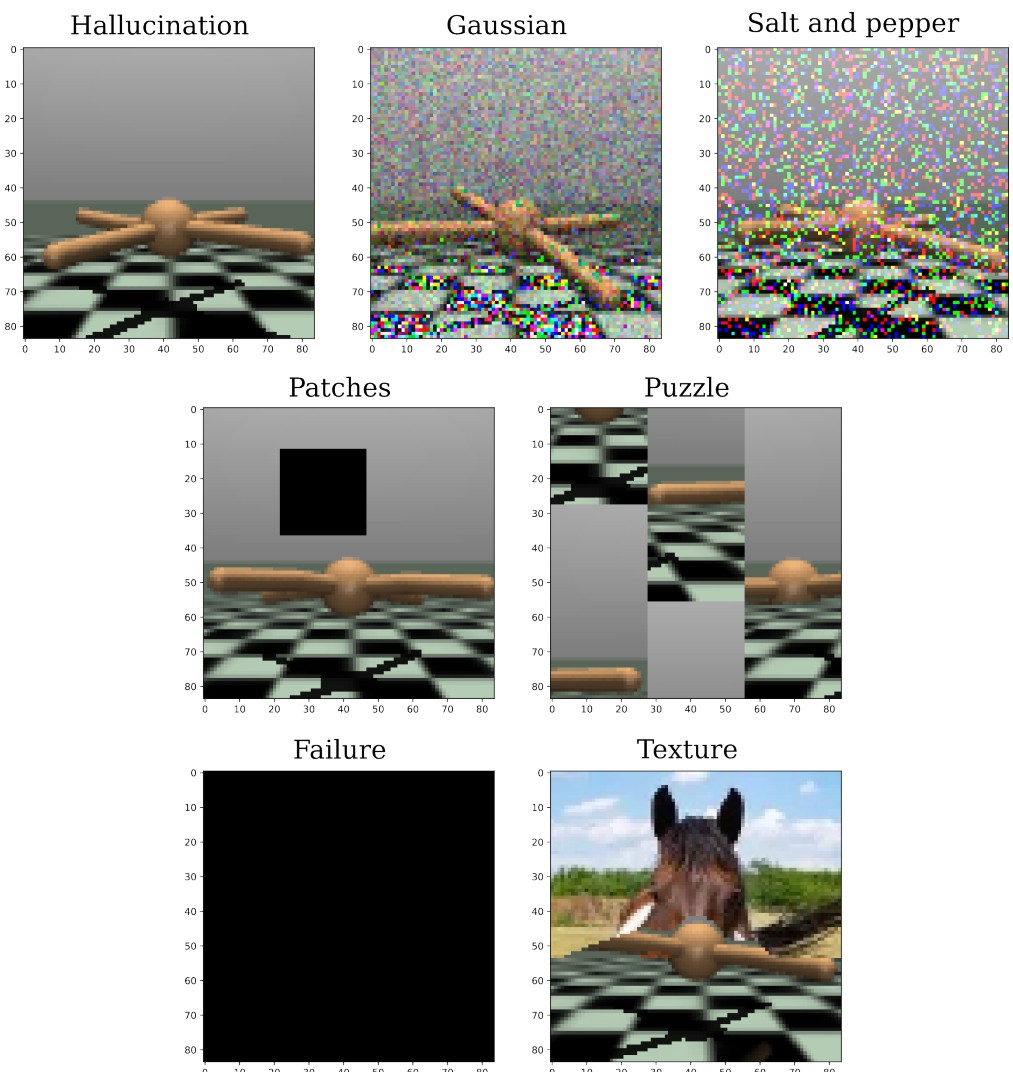

Figure 7: Examples of noises on the RGB images for the `MuJoCo`Ant-v5 environment.

## A.3 ADDITIONAL MUJOCO RESULTS

In Figures 8 and 9, we provide additional results on the `MuJoCo`environment. They illustrate the average reward and the standard deviation for every combination of noise for all the environments in the suite. Noise is applied to 1 modality at a time. Results are obtained from 50 evaluation trajectories after the models have fully trained.

## A.4 ADDITIONAL FETCH RESULTS

Below, we provide additional results on the `Fetch`environment. The tables describe the average reward and the standard deviation for every combination of noise for the Pick and Place and the Slide environments. Noise is applied to either 1 or 2 modalities at a time. Results are obtained from 50 evaluation trajectories after the models have fully trained.

Table 5: Return of multimodal fusion methods under Failure corruptions applied to one modality on `Fetch` – PickAndPlace, for increasing corruption probabilities.

| Model | 0.1 | 0.25 | 0.5 | 0.75 | 0.9 | 0.99 |
|---|---|---|---|---|---|---|
| Linear Comb | -0.82 ± 0.16 | -1.2 ± 1.38 | -1.65 ± 2.14 | -1.53 ± 1.68 | -1.68 ± 0.62 | -1.37 ± 0.76 |
| ConCat | 0.8 ± 1.14 | 0.95 ± 1.59 | 0.44 ± 0.84 | -0.49 ± 0.5 | -0.17 ± 0.37 | -0.92 ± 0.47 |
| CURL | 1.58 ± 1.0 | 1.08 ± 1.11 | -1.56 ± 1.0 | -2.08 ± 0.2 | -2.29 ± 0.46 | -2.56 ± 1.36 |
| GMC | -0.01 ± 1.15 | -1.34 ± 0.87 | -1.51 ± 0.77 | -1.62 ± 0.63 | -1.63 ± 0.54 | -1.6 ± 0.57 |
| AMDF | **2.73 ± 1.11** | 2.09 ± 1.08 | 1.02 ± 0.93 | -0.3 ± 0.56 | -0.83 ± 1.98 | -1.52 ± 1.55 |
| CORAL | -0.81 ± 0.14 | -0.84 ± 0.12 | -0.85 ± 0.1 | -0.87 ± 0.26 | -1.04 ± 0.38 | -1.11 ± 0.56 |
| MetricMM | 2.31 ± 1.34 | **2.35 ± 1.08** | **2.03 ± 0.68** | **2.35 ± 1.18** | **2.21 ± 1.2** | **1.89 ± 1.2** |

Table 6: Return of multimodal fusion methods under Gaussian corruptions applied to one modality on `Fetch` – PickAndPlace, for increasing corruption probabilities.

| Model | 0.1 | 0.25 | 0.5 | 0.75 | 0.9 | 0.99 |
|---|---|---|---|---|---|---|
| LinearComb | -0.84 ± 0.54 | -1.02 ± 0.32 | -0.78 ± 0.64 | -1.37 ± 0.8 | -1.42 ± 0.85 | -1.36 ± 1.07 |
| ConCat | 1.36 ± 1.06 | 0.28 ± 0.36 | -0.39 ± 0.92 | -1.89 ± 0.44 | -1.68 ± 1.11 | -1.51 ± 0.66 |
| CURL | **2.97 ± 1.32** | 1.33 ± 1.23 | 0.13 ± 0.99 | -0.75 ± 0.41 | -1.11 ± 0.41 | -1.43 ± 0.16 |
| GMC | -0.24 ± 1.86 | -0.15 ± 0.78 | -0.55 ± 0.71 | -1.42 ± 0.9 | -1.94 ± 1.77 | -0.85 ± 0.4 |
| AMDF | 2.57 ± 1.69 | 1.97 ± 0.94 | 1.07 ± 1.41 | -0.04 ± 1.08 | -0.24 ± 0.35 | -0.56 ± 0.28 |
| CORAL | -0.36 ± 0.59 | -0.58 ± 0.23 | -0.57 ± 0.24 | -0.78 ± 0.13 | -0.61 ± 0.19 | -0.5 ± 0.38 |
| MetricMM | 2.29 ± 1.3 | **2.52 ± 1.67** | **2.28 ± 0.87** | **1.97 ± 1.33** | **1.95 ± 1.17** | **2.19 ± 1.2** |

Table 7: Return of multimodal fusion methods under Hallucination corruptions applied to one modality on `Fetch` – PickAndPlace, for increasing corruption probabilities.

| Model | 0.1 | 0.25 | 0.5 | 0.75 | 0.9 | 0.99 |
|---|---|---|---|---|---|---|
| LinearComb | -0.95 ± 0.21 | -1.44 ± 0.58 | -1.28 ± 0.33 | -1.57 ± 0.53 | -1.82 ± 0.7 | -1.93 ± 0.81 |
| ConCat | -0.08 ± 0.93 | -0.77 ± 0.47 | -1.84 ± 0.99 | -2.38 ± 0.46 | -2.67 ± 0.6 | -2.44 ± 1.27 |
| CURL | 2.26 ± 0.81 | **2.44 ± 0.65** | 0.69 ± 0.3 | 0.32 ± 0.26 | -0.48 ± 0.84 | -0.83 ± 0.48 |
| GMC | 0.28 ± 1.29 | 0.12 ± 0.81 | -1.72 ± 1.39 | -1.42 ± 1.17 | -1.37 ± 0.82 | -1.15 ± 0.64 |
| AMDF | **2.43 ± 1.19** | 1.03 ± 0.78 | 0.25 ± 0.34 | -0.41 ± 0.7 | -1.36 ± 0.55 | -1.21 ± 0.16 |
| CORAL | -0.43 ± 0.52 | -0.58 ± 0.3 | -0.59 ± 0.33 | -1.02 ± 0.21 | -1.24 ± 0.35 | -0.96 ± 0.13 |
| MetricMM | 2.17 ± 0.73 | 1.99 ± 1.23 | **1.85 ± 1.09** | **1.85 ± 0.96** | **2.11 ± 0.84** | **2.08 ± 1.05** |

Table 8: Return of multimodal fusion methods under Patches corruptions applied to one modality on `Fetch` – PickAndPlace, for increasing corruption probabilities.

| Model | 0.1 | 0.25 | 0.5 | 0.75 | 0.9 | 0.99 |
|---|---|---|---|---|---|---|
| LinearComb | -0.75 ± 0.57 | -1.07 ± 0.87 | -1.42 ± 1.13 | -1.36 ± 0.94 | -2.13 ± 0.97 | -2.59 ± 1.68 |
| ConCat | 0.14 ± 0.18 | -0.55 ± 0.29 | -1.73 ± 1.0 | -1.62 ± 0.64 | -1.47 ± 1.11 | -1.97 ± 0.58 |
| CURL | 1.52 ± 0.83 | -0.56 ± 1.15 | -2.43 ± 0.93 | -2.8 ± 1.53 | -3.14 ± 1.16 | -2.67 ± 0.32 |
| GMC | -0.1 ± 0.43 | -1.78 ± 0.78 | -1.8 ± 0.94 | -1.37 ± 0.7 | -1.92 ± 0.92 | -1.28 ± 0.41 |
| AMDF | **2.76 ± 1.44** | 1.28 ± 1.03 | 0.56 ± 0.89 | -1.3 ± 0.61 | -1.09 ± 0.37 | -1.98 ± 1.28 |
| CORAL | -0.59 ± 0.19 | -1.22 ± 0.73 | -1.26 ± 0.77 | -1.23 ± 0.75 | -1.5 ± 1.18 | -1.13 ± 0.62 |
| MetricMM | 2.33 ± 1.16 | **2.13 ± 1.32** | **1.89 ± 1.21** | **2.22 ± 1.51** | **2.13 ± 1.39** | **2.23 ± 1.56** |

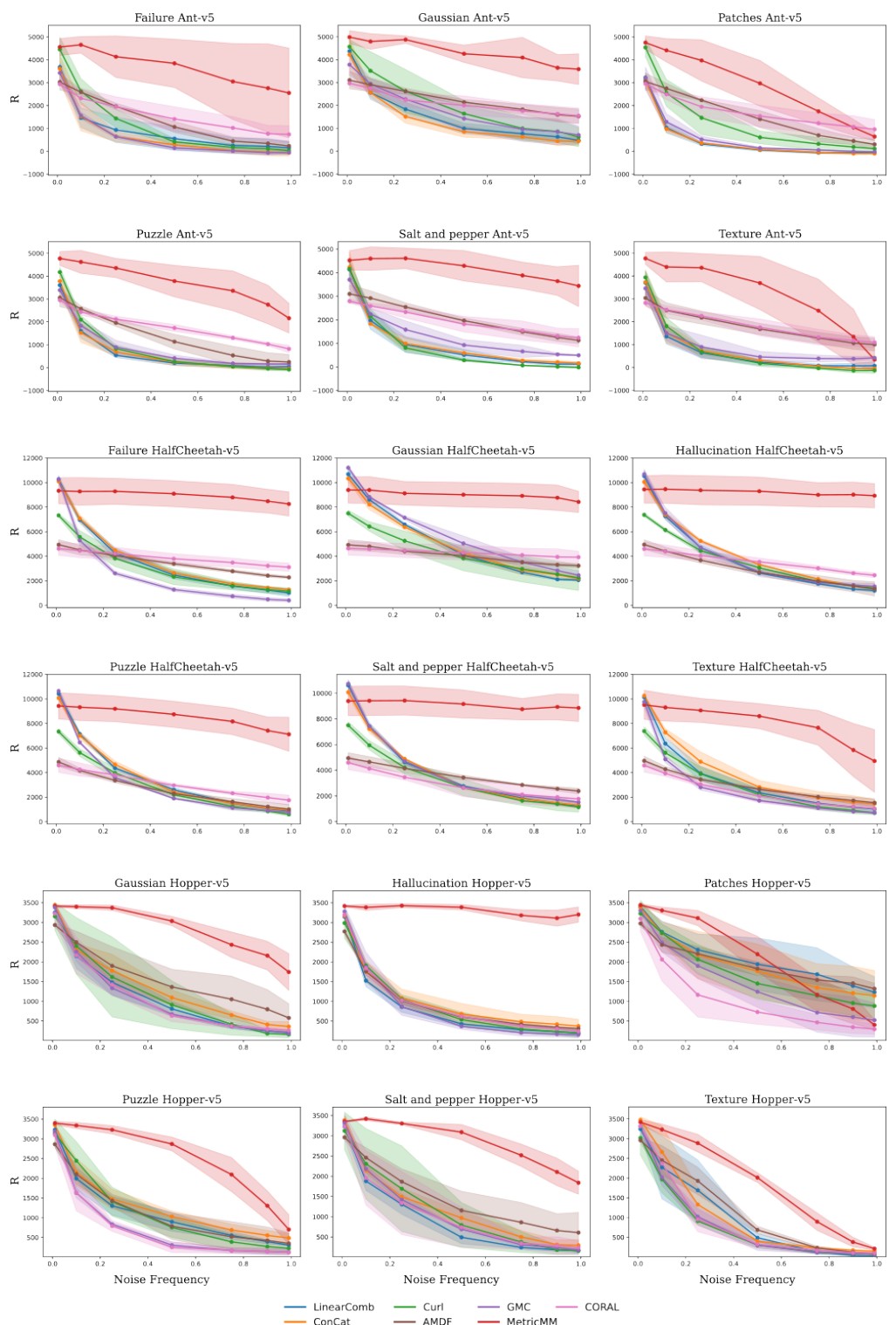

Figure 8: Mean and standard deviation over 5 seeds and 50 trajectories of the testing return for a SAC agent on the Mujoco suite. The policies are tested with different state estimation modules and different amounts of noise (perturbations of one modality at a time).

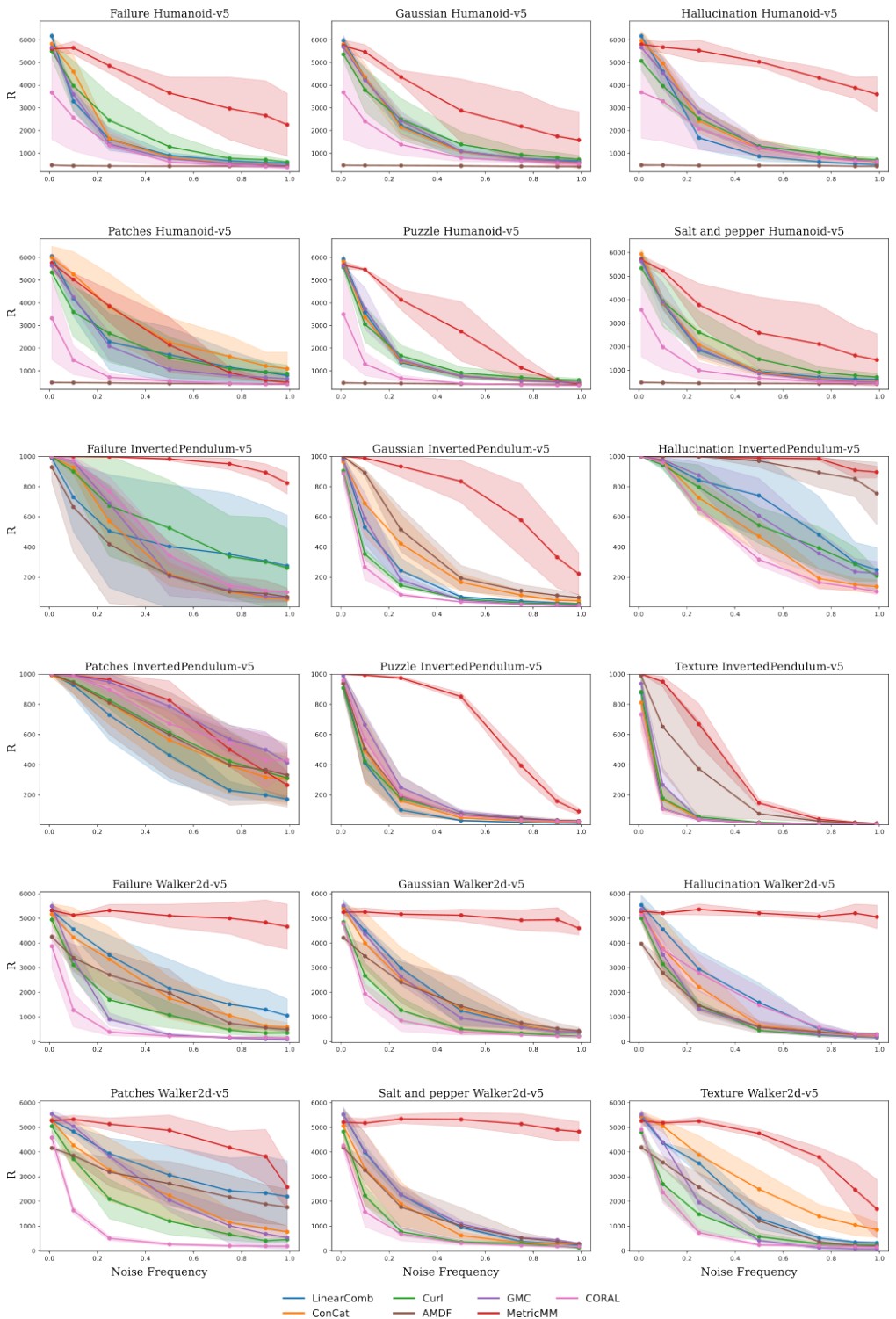

Figure 9: Mean and standard deviation over 5 seeds and 50 trajectories of the testing return for a SAC agent on the Mujoco suite. The policies are tested with different state estimation modules and different amounts of noise (perturbations of one modality at a time).

Table 9: Return of multimodal fusion methods under Puzzle corruptions applied to one modality on `Fetch` – PickAndPlace, for increasing corruption probabilities.

| Model | 0.1 | 0.25 | 0.5 | 0.75 | 0.9 | 0.99 |
|---|---|---|---|---|---|---|
| LinearComb | -0.29 ± 0.4 | -0.5 ± 0.58 | -1.39 ± 0.47 | -1.23 ± 0.74 | -1.03 ± 0.22 | -1.66 ± 0.78 |
| ConCat | 0.64 ± 1.05 | -0.25 ± 1.48 | -0.97 ± 0.21 | -1.18 ± 0.52 | -2.62 ± 1.45 | -1.71 ± 0.31 |
| CURL | **2.79 ± 1.14** | **2.79 ± 1.7** | 1.02 ± 0.19 | -0.83 ± 0.47 | -1.69 ± 0.63 | -2.26 ± 0.32 |
| GMC | -0.87 ± 1.02 | -0.97 ± 0.55 | -1.54 ± 0.61 | -2.23 ± 1.3 | -1.31 ± 0.42 | -1.4 ± 0.49 |
| AMDF | 2.4 ± 1.67 | 1.69 ± 1.14 | -0.67 ± 1.79 | -2.02 ± 0.47 | -2.61 ± 0.6 | -3.89 ± 1.4 |
| CORAL | -0.55 ± 0.32 | -0.64 ± 0.11 | -1.36 ± 0.93 | -1.38 ± 0.65 | -1.53 ± 0.99 | -1.39 ± 0.62 |
| MetricMM | 2.24 ± 0.89 | 2.26 ± 0.7 | **1.71 ± 1.27** | **2.37 ± 1.33** | **1.85 ± 1.4** | **1.98 ± 1.39** |

Table 10: Return of multimodal fusion methods under Salt and Pepper corruptions applied to one modality on `Fetch` – PickAndPlace, for increasing corruption probabilities.

| Model | 0.1 | 0.25 | 0.5 | 0.75 | 0.9 | 0.99 |
|---|---|---|---|---|---|---|
| LinearComb | -0.33 ± 0.56 | -0.68 ± 0.19 | -0.68 ± 0.1 | -0.83 ± 0.02 | -1.08 ± 0.22 | -0.72 ± 0.1 |
| ConCat | 0.56 ± 1.5 | 1.08 ± 0.95 | 0.62 ± 0.67 | -0.76 ± 1.26 | -0.79 ± 0.56 | -0.89 ± 0.43 |
| CURL | **3.56 ± 0.93** | **2.33 ± 0.88** | 1.64 ± 0.85 | 0.49 ± 0.97 | 0.15 ± 1.29 | -0.66 ± 0.85 |
| GMC | -0.15 ± 0.96 | -0.92 ± 1.3 | -1.64 ± 1.06 | -1.24 ± 0.98 | -1.9 ± 1.1 | -1.43 ± 0.57 |
| AMDF | 2.54 ± 1.07 | 1.83 ± 0.7 | **1.68 ± 0.55** | -0.12 ± 0.23 | 0.01 ± 0.09 | -0.83 ± 0.43 |
| CORAL | -0.39 ± 0.48 | -0.53 ± 0.3 | -0.62 ± 0.16 | -0.83 ± 0.1 | -0.88 ± 0.12 | -0.83 ± 0.1 |
| MetricMM | 2.43 ± 1.61 | 1.83 ± 1.22 | 1.67 ± 0.51 | **2.21 ± 1.24** | **1.89 ± 0.58** | **2.26 ± 1.04** |

Table 11: Return of multimodal fusion methods under Texture corruptions applied to one modality on `Fetch` – PickAndPlace, for increasing corruption probabilities.

| Model | 0.1 | 0.25 | 0.5 | 0.75 | 0.9 | 0.99 |
|---|---|---|---|---|---|---|
| LinearComb | -0.22 ± 0.97 | -0.63 ± 0.76 | -0.42 ± 1.38 | -0.55 ± 0.5 | -0.36 ± 0.41 | -0.02 ± 0.69 |
| ConCat | 1.56 ± 0.78 | 0.89 ± 0.96 | 0.71 ± 1.69 | -0.14 ± 1.36 | -0.62 ± 2.25 | 0.86 ± 1.91 |
| CURL | 2.39 ± 1.46 | 0.09 ± 1.02 | -1.97 ± 0.38 | -2.9 ± 0.35 | -2.89 ± 0.2 | -3.2 ± 0.31 |
| GMC | -0.1 ± 0.71 | -1.12 ± 0.86 | -1.41 ± 1.45 | -2.02 ± 1.33 | -1.6 ± 1.01 | -1.65 ± 0.97 |
| AMDF | **3.21 ± 2.03** | **2.36 ± 1.66** | 1.26 ± 1.34 | 0.1 ± 0.67 | -0.79 ± 0.19 | 0.09 ± 0.22 |
| CORAL | -0.35 ± 0.54 | -0.67 ± 0.14 | -0.99 ± 0.3 | -1.8 ± 1.38 | -1.54 ± 1.08 | -1.09 ± 0.41 |
| MetricMM | 1.84 ± 0.94 | 2.1 ± 1.09 | **2.52 ± 1.36** | **2.19 ± 1.01** | **2.27 ± 1.22** | **1.95 ± 1.31** |

Table 12: Return of multimodal fusion methods under Failure corruptions applied simultaneously to two modalities on `Fetch` – PickAndPlace, for increasing corruption probabilities.

| Model | 0.1 | 0.25 | 0.5 | 0.75 | 0.9 | 0.99 |
|---|---|---|---|---|---|---|
| LinearComb | -1.32 ± 1.36 | -1.54 ± 0.82 | -1.36 ± 0.54 | -2.28 ± 1.32 | -2.04 ± 1.22 | -2.37 ± 1.16 |
| ConCat | 0.17 ± 1.29 | -0.25 ± 0.1 | -1.17 ± 0.79 | -2.37 ± 1.99 | -2.1 ± 0.19 | -2.78 ± 0.99 |
| CURL | 1.32 ± 0.43 | -0.15 ± 1.5 | -2.39 ± 0.83 | -2.27 ± 0.77 | -2.55 ± 0.65 | -2.64 ± 0.72 |
| GMC | 0.05 ± 1.06 | -1.64 ± 0.63 | -1.68 ± 0.59 | -2.18 ± 0.38 | -2.15 ± 0.47 | -2.28 ± 0.45 |
| AMDF | **2.76 ± 1.39** | 1.93 ± 1.72 | -0.18 ± 1.06 | -1.27 ± 0.87 | -2.39 ± 1.21 | **-1.77 ± 0.59** |
| CORAL | -0.6 ± 0.14 | -1.09 ± 0.43 | -1.1 ± 0.53 | -1.34 ± 0.82 | **-1.35 ± 0.81** | -1.83 ± 1.51 |
| MetricMM | 2.22 ± 1.74 | **2.44 ± 1.48** | **1.65 ± 0.64** | **-0.47 ± 1.2** | -1.51 ± 1.25 | -3.0 ± 2.27 |

Table 13: Return of multimodal fusion methods under Gaussian corruptions applied simultaneously to two modalities on `Fetch` – PickAndPlace, for increasing corruption probabilities.

| Model | 0.1 | 0.25 | 0.5 | 0.75 | 0.9 | 0.99 |
|---|---|---|---|---|---|---|
| LinearComb | -0.54 ± 0.67 | -0.99 ± 0.54 | -1.64 ± 0.8 | -1.21 ± 0.31 | -1.91 ± 0.78 | -1.38 ± 0.36 |
| ConCat | -0.05 ± 0.58 | -1.05 ± 0.66 | -1.96 ± 1.91 | -2.63 ± 0.47 | -3.09 ± 0.98 | -3.27 ± 0.23 |
| CURL | 1.89 ± 0.74 | 0.29 ± 0.6 | -1.5 ± 0.81 | -1.94 ± 0.24 | -2.11 ± 0.47 | -2.04 ± 0.54 |
| GMC | -0.1 ± 1.09 | -0.88 ± 0.91 | -1.59 ± 1.1 | -1.57 ± 1.08 | -1.22 ± 0.44 | -1.4 ± 0.52 |
| AMDF | **2.21 ± 1.56** | 1.56 ± 1.24 | -0.27 ± 0.45 | -2.15 ± 0.69 | -2.65 ± 0.96 | -2.64 ± 0.5 |
| CORAL | -0.48 ± 0.38 | -0.59 ± 0.25 | -0.92 ± 0.25 | -0.81 ± 0.03 | -1.1 ± 0.44 | **-0.88 ± 0.12** |
| MetricMM | 2.1 ± 1.49 | **2.46 ± 1.23** | **1.76 ± 1.5** | **0.46 ± 0.63** | **-0.83 ± 0.39** | -2.4 ± 0.5 |

Table 14: Return of multimodal fusion methods under Hallucination corruptions applied simultaneously to two modalities on `Fetch` – PickAndPlace, for increasing corruption probabilities.

| Model | 0.1 | 0.25 | 0.5 | 0.75 | 0.9 | 0.99 |
|---|---|---|---|---|---|---|
| LinearComb | -0.24 ± 0.61 | -1.4 ± 0.88 | -1.73 ± 0.65 | -1.48 ± 0.48 | -1.92 ± 1.1 | **-1.81 ± 0.74** |
| ConCat | -0.13 ± 0.7 | -0.92 ± 1.23 | -2.55 ± 0.79 | -2.69 ± 0.62 | -3.52 ± 0.62 | -3.95 ± 0.83 |
| CURL | 2.19 ± 0.59 | 0.43 ± 0.66 | -0.76 ± 0.36 | -2.22 ± 0.1 | -2.24 ± 0.24 | -2.83 ± 0.32 |
| GMC | -0.12 ± 0.58 | -1.31 ± 0.68 | -1.36 ± 0.73 | -2.01 ± 1.02 | -2.26 ± 1.15 | -2.88 ± 1.63 |
| AMDF | 1.43 ± 0.91 | -0.2 ± 0.53 | -0.87 ± 0.69 | -1.79 ± 0.38 | -2.85 ± 0.23 | -2.75 ± 0.23 |
| CORAL | -0.6 ± 0.23 | -0.92 ± 0.38 | -1.32 ± 0.46 | -1.46 ± 0.54 | -1.72 ± 0.8 | -2.25 ± 1.28 |
| MetricMM | **2.36 ± 1.18** | **2.13 ± 1.33** | **1.86 ± 1.07** | **0.71 ± 0.46** | **-0.87 ± 0.54** | -3.46 ± 1.15 |

Table 15: Return of multimodal fusion methods under Puzzle corruptions applied simultaneously to two modalities on `Fetch` – PickAndPlace, for increasing corruption probabilities.

| Model | 0.1 | 0.25 | 0.5 | 0.75 | 0.9 | 0.99 |
|---|---|---|---|---|---|---|
| LinearComb | -0.77 ± 0.81 | -1.09 ± 0.35 | -1.07 ± 0.32 | -1.97 ± 0.9 | -1.74 ± 0.72 | -1.63 ± 0.61 |
| ConCat | 0.09 ± 2.02 | -1.83 ± 1.13 | -2.94 ± 1.32 | -2.6 ± 0.65 | -1.92 ± 0.51 | -2.01 ± 0.67 |
| CURL | **2.61 ± 1.13** | 1.34 ± 0.86 | -1.41 ± 0.47 | -1.8 ± 0.77 | -1.77 ± 0.03 | **-1.35 ± 0.0** |
| GMC | -1.15 ± 1.29 | -1.27 ± 0.52 | -2.43 ± 1.63 | -1.37 ± 0.4 | -1.53 ± 0.52 | -2.22 ± 1.2 |
| AMDF | 1.94 ± 1.22 | -0.04 ± 0.64 | -1.9 ± 1.32 | -2.4 ± 0.6 | -2.38 ± 0.78 | -2.31 ± 0.74 |
| CORAL | -0.34 ± 0.56 | -1.12 ± 0.51 | -1.22 ± 0.52 | -1.49 ± 0.67 | -1.26 ± 0.3 | -1.36 ± 0.39 |
| MetricMM | 1.94 ± 0.51 | **2.18 ± 1.01** | **1.79 ± 1.64** | **0.38 ± 0.69** | **-1.05 ± 0.49** | -1.86 ± 0.71 |

Table 16: Return of multimodal fusion methods under Salt and Pepper corruptions applied simultaneously to two modalities on `Fetch` – PickAndPlace, for increasing corruption probabilities.

| Model | 0.1 | 0.25 | 0.5 | 0.75 | 0.9 | 0.99 |
|---|---|---|---|---|---|---|
| LinearComb | -0.55 ± 0.37 | -0.57 ± 0.2 | -1.03 ± 0.36 | -1.21 ± 0.46 | -0.99 ± 0.2 | -0.96 ± 0.16 |
| ConCat | -0.02 ± 1.22 | 0.25 ± 1.43 | -0.91 ± 0.99 | -2.06 ± 1.24 | -1.94 ± 0.51 | -1.38 ± 1.37 |
| CURL | **2.53 ± 0.66** | **2.08 ± 0.79** | 0.12 ± 1.15 | -0.7 ± 0.28 | -1.17 ± 0.58 | -1.26 ± 0.7 |
| GMC | -0.45 ± 0.34 | -1.34 ± 0.76 | -1.34 ± 0.4 | -1.27 ± 0.53 | -1.42 ± 0.48 | -1.72 ± 0.92 |
| AMDF | 2.22 ± 1.1 | 0.46 ± 0.21 | 0.17 ± 0.26 | -0.73 ± 0.34 | -1.31 ± 0.36 | -1.38 ± 0.26 |
| CORAL | -0.56 ± 0.26 | -0.56 ± 0.19 | -0.66 ± 0.08 | -0.83 ± 0.1 | -1.25 ± 0.61 | **-0.83 ± 0.12** |
| MetricMM | 2.24 ± 0.62 | 1.78 ± 1.07 | **1.59 ± 0.88** | **0.27 ± 0.6** | **-0.8 ± 0.83** | -1.54 ± 0.26 |

Table 17: Return of multimodal fusion methods under Failure corruptions applied to one modality on `Fetch` − Slide, for increasing corruption probabilities.

| Model | 0.1 | 0.25 | 0.5 | 0.75 | 0.9 | 0.99 |
|---|---|---|---|---|---|---|
| LinearComb | 6.36 ± 0.71 | 4.79 ± 2.13 | 4.58 ± 0.47 | 1.15 ± 1.23 | 0.5 ± 1.76 | -1.14 ± 2.41 |
| ConCat | 6.5 ± 0.17 | 5.49 ± 0.12 | 3.49 ± 1.86 | 2.8 ± 0.96 | 2.48 ± 2.0 | 1.53 ± 2.91 |
| CURL | 6.95 ± 0.95 | 7.41 ± 0.82 | 5.68 ± 0.92 | 2.72 ± 1.97 | 1.27 ± 1.93 | 0.16 ± 3.09 |
| GMC | 5.65 ± 1.48 | 4.39 ± 1.92 | 1.6 ± 2.39 | -0.42 ± 1.81 | -1.53 ± 2.62 | -1.76 ± 1.31 |
| AMDF | 4.03 ± 1.96 | 2.67 ± 1.64 | 2.1 ± 1.47 | -0.21 ± 0.58 | 0.17 ± 0.51 | 0.1 ± 1.53 |
| CORAL | 4.97 ± 3.43 | 4.63 ± 2.64 | 0.79 ± 0.66 | -0.35 ± 1.21 | -1.47 ± 0.48 | -1.32 ± 0.67 |
| MetricMM | **8.98 ± 1.49** | **9.31 ± 0.84** | **9.12 ± 0.51** | **8.25 ± 1.1** | **7.92 ± 1.29** | **8.46 ± 0.92** |

Table 18: Return of multimodal fusion methods under Gaussian corruptions applied to one modality on `Fetch` − Slide, for increasing corruption probabilities.

| Model | 0.1 | 0.25 | 0.5 | 0.75 | 0.9 | 0.99 |
|---|---|---|---|---|---|---|
| LinearComb | 6.96 ± 1.16 | 6.55 ± 1.46 | 4.02 ± 0.4 | 3.64 ± 1.64 | 3.6 ± 1.24 | 2.17 ± 2.3 |
| ConCat | 6.33 ± 0.36 | 6.74 ± 1.48 | 4.79 ± 1.24 | 4.76 ± 1.49 | 4.41 ± 2.19 | 4.35 ± 2.38 |
| CURL | 9.11 ± 0.72 | 6.46 ± 0.86 | 4.92 ± 1.53 | 3.29 ± 0.45 | 2.57 ± 1.91 | 1.77 ± 0.86 |
| GMC | 6.88 ± 0.92 | 4.33 ± 0.25 | 2.41 ± 0.78 | 0.97 ± 0.71 | -1.24 ± 1.5 | -0.81 ± 0.99 |
| AMDF | 4.56 ± 2.47 | 2.7 ± 1.05 | 1.65 ± 0.82 | 0.32 ± 0.43 | 0.26 ± 1.18 | -1.02 ± 0.69 |
| CORAL | 5.25 ± 3.29 | 4.04 ± 2.01 | 2.88 ± 2.27 | 1.59 ± 0.32 | -0.16 ± 1.25 | -0.24 ± 0.74 |
| MetricMM | **9.34 ± 0.61** | **9.54 ± 1.26** | **9.7 ± 0.38** | **8.14 ± 1.14** | **8.99 ± 1.35** | **8.28 ± 1.01** |

Table 19: Return of multimodal fusion methods under Hallucination corruptions applied to one modality on `Fetch` − Slide, for increasing corruption probabilities.

| Model | 0.1 | 0.25 | 0.5 | 0.75 | 0.9 | 0.99 |
|---|---|---|---|---|---|---|
| LinearComb | 6.76 ± 0.67 | 6.27 ± 0.91 | 5.78 ± 0.72 | 5.62 ± 0.71 | 4.21 ± 0.44 | 4.16 ± 1.37 |
| ConCat | 7.14 ± 0.05 | 6.61 ± 1.72 | 5.57 ± 1.23 | 5.21 ± 1.12 | 4.04 ± 0.99 | 2.57 ± 1.11 |
| CURL | 8.96 ± 0.8 | 7.99 ± 1.5 | 7.58 ± 0.9 | 5.96 ± 0.64 | 4.87 ± 0.87 | 4.72 ± 0.75 |
| GMC | 7.07 ± 1.01 | 6.53 ± 0.72 | 3.9 ± 1.0 | 2.95 ± 0.82 | 3.53 ± 0.83 | 2.71 ± 0.09 |
| AMDF | 3.62 ± 1.73 | 3.75 ± 1.69 | 4.06 ± 1.92 | 2.28 ± 1.17 | 2.25 ± 0.99 | 1.62 ± 1.18 |
| CORAL | 5.72 ± 3.19 | 5.13 ± 2.82 | 4.02 ± 3.01 | 3.54 ± 2.7 | 2.62 ± 1.68 | 2.28 ± 2.19 |
| MetricMM | **9.2 ± 1.15** | **8.31 ± 0.56** | **8.58 ± 0.57** | **9.27 ± 0.88** | **9.18 ± 0.45** | **8.48 ± 0.34** |

Table 20: Return of multimodal fusion methods under Patches corruptions applied to one modality on `Fetch` − Slide, for increasing corruption probabilities.

| Model | 0.1 | 0.25 | 0.5 | 0.75 | 0.9 | 0.99 |
|---|---|---|---|---|---|---|
| LinearComb | 5.56 ± 0.6 | 4.16 ± 0.98 | 2.81 ± 1.12 | 1.96 ± 2.34 | 0.35 ± 1.86 | -0.36 ± 1.16 |
| ConCat | 7.5 ± 0.94 | 4.7 ± 0.56 | 3.54 ± 0.9 | 2.75 ± 1.69 | 1.3 ± 2.3 | 1.3 ± 2.44 |
| CURL | 8.42 ± 1.34 | 5.95 ± 1.43 | 4.53 ± 1.76 | 1.41 ± 4.01 | -0.01 ± 3.13 | -0.68 ± 4.36 |
| GMC | 5.39 ± 0.65 | 3.49 ± 1.84 | -0.25 ± 2.09 | -2.62 ± 2.24 | -1.57 ± 1.11 | -3.09 ± 1.44 |
| AMDF | 3.82 ± 2.01 | 3.29 ± 1.49 | 0.87 ± 0.41 | -0.06 ± 1.41 | -0.62 ± 1.0 | -0.4 ± 1.7 |
| CORAL | 3.94 ± 2.84 | 2.08 ± 1.11 | -0.88 ± 0.42 | -0.92 ± 1.4 | -2.23 ± 1.46 | -1.85 ± 0.41 |
| MetricMM | **8.75 ± 0.7** | **9.19 ± 1.06** | **9.37 ± 0.95** | **8.13 ± 0.09** | **7.26 ± 1.01** | **8.51 ± 0.79** |

Table 21: Return of multimodal fusion methods under Puzzle corruptions applied to one modality on `Fetch` − Slide, for increasing corruption probabilities.

| Model | 0.1 | 0.25 | 0.5 | 0.75 | 0.9 | 0.99 |
|---|---|---|---|---|---|---|
| LinearComb | $6.83 \pm 1.01$ | $5.47 \pm 0.87$ | $4.65 \pm 1.06$ | $0.7 \pm 0.69$ | $-0.18 \pm 0.6$ | $-1.47 \pm 1.56$ |
| ConCat | $6.35 \pm 1.87$ | $5.95 \pm 1.73$ | $4.38 \pm 1.99$ | $2.26 \pm 2.15$ | $1.56 \pm 2.42$ | $-0.05 \pm 2.35$ |
| CURL | $8.44 \pm 0.25$ | $6.84 \pm 0.53$ | $5.42 \pm 1.47$ | $3.55 \pm 2.58$ | $1.13 \pm 2.33$ | $0.59 \pm 3.19$ |
| GMC | $5.55 \pm 1.65$ | $4.7 \pm 1.86$ | $1.29 \pm 1.77$ | $-1.75 \pm 1.02$ | $-3.12 \pm 1.66$ | $-2.94 \pm 1.66$ |
| AMDF | $3.88 \pm 1.79$ | $3.11 \pm 1.55$ | $2.32 \pm 1.47$ | $0.51 \pm 0.97$ | $-0.85 \pm 1.75$ | $-0.89 \pm 1.61$ |
| CORAL | $4.95 \pm 2.95$ | $3.87 \pm 2.33$ | $0.56 \pm 1.75$ | $-1.41 \pm 1.09$ | $-2.0 \pm 0.96$ | $-1.81 \pm 0.57$ |
| MetricMM | $\mathbf{8.69 \pm 0.62}$ | $\mathbf{9.44 \pm 0.44}$ | $\mathbf{9.19 \pm 1.29}$ | $\mathbf{7.97 \pm 0.3}$ | $\mathbf{8.55 \pm 0.82}$ | $\mathbf{7.9 \pm 1.44}$ |

Table 22: Return of multimodal fusion methods under Salt and Pepper corruptions applied to one modality on `Fetch` − Slide, for increasing corruption probabilities.

| Model | 0.1 | 0.25 | 0.5 | 0.75 | 0.9 | 0.99 |
|---|---|---|---|---|---|---|
| LinearComb | $6.48 \pm 0.14$ | $5.76 \pm 0.24$ | $5.64 \pm 1.01$ | $5.34 \pm 0.81$ | $4.82 \pm 1.31$ | $3.37 \pm 1.21$ |
| ConCat | $6.41 \pm 1.52$ | $6.05 \pm 1.48$ | $6.48 \pm 1.03$ | $5.18 \pm 1.59$ | $3.49 \pm 2.32$ | $3.6 \pm 2.02$ |
| CURL | $8.55 \pm 1.54$ | $7.14 \pm 0.76$ | $6.51 \pm 0.27$ | $5.73 \pm 1.55$ | $3.95 \pm 0.79$ | $3.15 \pm 0.85$ |
| GMC | $7.34 \pm 1.55$ | $6.36 \pm 0.85$ | $4.87 \pm 1.99$ | $4.21 \pm 2.5$ | $2.46 \pm 1.51$ | $1.16 \pm 2.65$ |
| AMDF | $4.13 \pm 2.24$ | $3.01 \pm 1.32$ | $2.85 \pm 1.1$ | $1.65 \pm 1.31$ | $0.91 \pm 0.47$ | $0.2 \pm 0.44$ |
| CORAL | $5.04 \pm 3.53$ | $4.3 \pm 2.26$ | $5.29 \pm 2.91$ | $4.61 \pm 2.6$ | $2.46 \pm 1.97$ | $2.33 \pm 1.59$ |
| MetricMM | $\mathbf{9.57 \pm 0.9}$ | $\mathbf{8.69 \pm 0.79}$ | $\mathbf{9.64 \pm 0.65}$ | $\mathbf{8.54 \pm 1.41}$ | $\mathbf{8.03 \pm 0.86}$ | $\mathbf{8.71 \pm 0.7}$ |

Table 23: Return of multimodal fusion methods under Texture corruptions applied to one modality on `Fetch` − Slide, for increasing corruption probabilities.

| Model | 0.1 | 0.25 | 0.5 | 0.75 | 0.9 | 0.99 |
|---|---|---|---|---|---|---|
| LinearComb | $6.54 \pm 0.73$ | $7.09 \pm 0.08$ | $4.63 \pm 1.19$ | $2.07 \pm 1.11$ | $2.69 \pm 2.55$ | $1.02 \pm 1.74$ |
| ConCat | $7.32 \pm 1.39$ | $5.64 \pm 1.38$ | $5.32 \pm 2.07$ | $4.93 \pm 2.86$ | $4.59 \pm 2.54$ | $3.52 \pm 3.85$ |
| CURL | $8.01 \pm 0.92$ | $5.71 \pm 0.92$ | $2.85 \pm 1.27$ | $-0.52 \pm 0.52$ | $-2.22 \pm 0.22$ | $-2.48 \pm 0.37$ |
| GMC | $6.36 \pm 0.8$ | $4.43 \pm 0.41$ | $1.9 \pm 1.58$ | $-1.27 \pm 1.59$ | $-2.15 \pm 2.31$ | $-2.65 \pm 0.57$ |
| AMDF | $3.28 \pm 1.83$ | $2.69 \pm 1.71$ | $1.03 \pm 0.35$ | $-0.65 \pm 0.52$ | $-1.5 \pm 0.77$ | $-1.83 \pm 0.68$ |
| CORAL | $4.49 \pm 2.76$ | $3.16 \pm 2.03$ | $1.71 \pm 1.34$ | $-1.02 \pm 0.28$ | $-1.75 \pm 0.79$ | $-2.43 \pm 0.86$ |
| MetricMM | $\mathbf{9.45 \pm 0.49}$ | $\mathbf{9.01 \pm 0.97}$ | $\mathbf{9.4 \pm 0.65}$ | $\mathbf{7.54 \pm 0.18}$ | $\mathbf{9.02 \pm 1.12}$ | $\mathbf{7.6 \pm 0.47}$ |

Table 24: Return of multimodal fusion methods under Gaussian corruptions applied simultaneously to two modalities on `Fetch` − Slide, for increasing corruption probabilities.

| Model | 0.1 | 0.25 | 0.5 | 0.75 | 0.9 | 0.99 |
|---|---|---|---|---|---|---|
| LinearComb | $6.65 \pm 1.03$ | $4.15 \pm 0.62$ | $3.28 \pm 0.66$ | $-0.25 \pm 1.3$ | $-0.15 \pm 1.23$ | $-1.22 \pm 1.08$ |
| ConCat | $6.05 \pm 0.65$ | $5.69 \pm 0.73$ | $4.63 \pm 1.14$ | $2.64 \pm 1.41$ | $0.87 \pm 1.13$ | $0.31 \pm 1.46$ |
| CURL | $7.06 \pm 0.71$ | $6.68 \pm 0.11$ | $3.93 \pm 0.39$ | $-0.08 \pm 1.13$ | $-1.12 \pm 1.51$ | $-1.46 \pm 0.77$ |
| GMC | $5.28 \pm 0.58$ | $3.67 \pm 1.57$ | $1.05 \pm 1.06$ | $-2.04 \pm 1.34$ | $-2.45 \pm 1.56$ | $-3.75 \pm 2.28$ |
| AMDF | $4.02 \pm 1.71$ | $2.14 \pm 0.89$ | $-0.11 \pm 0.61$ | $-1.1 \pm 1.56$ | $-2.45 \pm 2.66$ | $-2.81 \pm 1.7$ |
| CORAL | $4.45 \pm 2.76$ | $3.0 \pm 1.74$ | $1.14 \pm 0.57$ | $-2.03 \pm 1.51$ | $-1.61 \pm 0.74$ | $-2.37 \pm 0.57$ |
| MetricMM | $\mathbf{10.1 \pm 0.61}$ | $\mathbf{8.71 \pm 0.56}$ | $\mathbf{7.15 \pm 0.76}$ | $\mathbf{6.13 \pm 1.63}$ | $\mathbf{4.38 \pm 2.15}$ | $\mathbf{2.47 \pm 1.89}$ |

Table 25: Return of multimodal fusion methods under Hallucination corruptions applied simultaneously to two modalities on `Fetch − Slide`, for increasing corruption probabilities.

| Model | 0.1 | 0.25 | 0.5 | 0.75 | 0.9 | 0.99 |
|---|---|---|---|---|---|---|
| LinearComb | $6.64 \pm 1.36$ | $5.88 \pm 0.37$ | $2.73 \pm 0.68$ | $1.51 \pm 0.36$ | $-0.01 \pm 0.3$ | $-0.71 \pm 0.4$ |
| ConCat | $6.68 \pm 1.49$ | $5.88 \pm 0.4$ | $3.37 \pm 1.53$ | $0.77 \pm 0.64$ | $-1.2 \pm 1.42$ | $-0.46 \pm 0.6$ |
| CURL | $8.23 \pm 0.94$ | $7.85 \pm 0.65$ | $4.08 \pm 0.57$ | $1.48 \pm 1.29$ | $-0.15 \pm 0.69$ | $0.21 \pm 0.68$ |
| GMC | $6.62 \pm 0.29$ | $4.99 \pm 1.0$ | $1.82 \pm 1.34$ | $0.03 \pm 0.39$ | $-1.02 \pm 1.24$ | $-1.86 \pm 0.47$ |
| AMDF | $3.74 \pm 1.24$ | $3.19 \pm 1.4$ | $1.71 \pm 0.48$ | $0.07 \pm 0.8$ | $-0.29 \pm 0.2$ | $-0.51 \pm 1.08$ |
| CORAL | $5.5 \pm 3.2$ | $4.39 \pm 2.65$ | $1.96 \pm 2.1$ | $-0.03 \pm 0.8$ | $-1.66 \pm 0.57$ | $-1.36 \pm 0.56$ |
| MetricMM | $\mathbf{8.67 \pm 1.48}$ | $\mathbf{8.88 \pm 0.55}$ | $\mathbf{8.15 \pm 1.19}$ | $\mathbf{6.94 \pm 0.87}$ | $\mathbf{5.94 \pm 0.97}$ | $\mathbf{3.32 \pm 1.65}$ |

Table 26: Return of multimodal fusion methods under Patches corruptions applied simultaneously to two modalities on `Fetch − Slide`, for increasing corruption probabilities.

| Model | 0.1 | 0.25 | 0.5 | 0.75 | 0.9 | 0.99 |
|---|---|---|---|---|---|---|
| LinearComb | $5.31 \pm 0.75$ | $5.32 \pm 1.59$ | $1.85 \pm 2.11$ | $-0.65 \pm 1.31$ | $-2.1 \pm 0.32$ | $-2.05 \pm 0.92$ |
| ConCat | $6.85 \pm 1.73$ | $5.96 \pm 1.41$ | $2.14 \pm 0.89$ | $-0.66 \pm 1.37$ | $-1.77 \pm 0.87$ | $-1.93 \pm 0.61$ |
| CURL | $8.31 \pm 1.27$ | $5.58 \pm 1.18$ | $2.58 \pm 2.41$ | $-1.27 \pm 0.75$ | $-2.74 \pm 1.79$ | $-2.16 \pm 1.2$ |
| GMC | $4.4 \pm 1.32$ | $2.45 \pm 1.09$ | $-0.65 \pm 1.59$ | $-1.02 \pm 0.49$ | $-2.23 \pm 1.04$ | $-1.97 \pm 0.22$ |
| AMDF | $3.24 \pm 1.94$ | $2.33 \pm 0.72$ | $0.74 \pm 1.77$ | $-1.37 \pm 1.86$ | $-2.06 \pm 2.96$ | $-2.54 \pm 2.7$ |
| CORAL | $3.69 \pm 1.61$ | $1.53 \pm 1.32$ | $-0.91 \pm 0.71$ | $-2.46 \pm 0.85$ | $-2.31 \pm 0.81$ | $-2.1 \pm 1.27$ |
| MetricMM | $\mathbf{8.62 \pm 0.73}$ | $\mathbf{7.81 \pm 0.74}$ | $\mathbf{4.03 \pm 0.74}$ | $\mathbf{1.74 \pm 1.77}$ | $\mathbf{-0.84 \pm 1.57}$ | $\mathbf{-0.12 \pm 1.46}$ |

Table 27: Return of multimodal fusion methods under Puzzle corruptions applied simultaneously to two modalities on `Fetch − Slide`, for increasing corruption probabilities.

| Model | 0.1 | 0.25 | 0.5 | 0.75 | 0.9 | 0.99 |
|---|---|---|---|---|---|---|
| LinearComb | $5.82 \pm 0.77$ | $5.18 \pm 0.76$ | $1.76 \pm 0.79$ | $-1.67 \pm 0.55$ | $-2.74 \pm 0.62$ | $-1.96 \pm 0.58$ |
| ConCat | $7.14 \pm 0.98$ | $5.31 \pm 1.1$ | $2.02 \pm 1.54$ | $0.5 \pm 2.03$ | $-1.9 \pm 1.54$ | $-2.73 \pm 1.48$ |
| CURL | $7.42 \pm 0.74$ | $5.5 \pm 0.57$ | $2.67 \pm 1.25$ | $0.96 \pm 0.69$ | $-0.27 \pm 0.66$ | $-0.76 \pm 0.7$ |
| GMC | $6.25 \pm 0.32$ | $2.82 \pm 1.22$ | $0.08 \pm 0.94$ | $-1.44 \pm 1.25$ | $-2.63 \pm 0.91$ | $-2.37 \pm 1.9$ |
| AMDF | $4.09 \pm 2.54$ | $3.36 \pm 2.0$ | $0.55 \pm 2.42$ | $-1.5 \pm 2.04$ | $-2.03 \pm 1.51$ | $-1.76 \pm 1.74$ |
| CORAL | $5.19 \pm 3.33$ | $3.62 \pm 1.92$ | $0.71 \pm 0.33$ | $-1.01 \pm 0.66$ | $-2.71 \pm 0.14$ | $-2.28 \pm 1.06$ |
| MetricMM | $\mathbf{9.12 \pm 0.87}$ | $\mathbf{7.05 \pm 1.52}$ | $\mathbf{6.17 \pm 0.06}$ | $\mathbf{3.79 \pm 0.77}$ | $\mathbf{1.43 \pm 0.58}$ | $\mathbf{1.21 \pm 1.53}$ |

Table 28: Return of multimodal fusion methods under Salt and Pepper corruptions applied simultaneously to two modalities on `Fetch − Slide`, for increasing corruption probabilities.

| Model | 0.1 | 0.25 | 0.5 | 0.75 | 0.9 | 0.99 |
|---|---|---|---|---|---|---|
| LinearComb | $6.3 \pm 0.8$ | $6.5 \pm 1.13$ | $5.02 \pm 1.15$ | $1.6 \pm 1.72$ | $-0.03 \pm 1.06$ | $0.79 \pm 1.85$ |
| ConCat | $5.56 \pm 1.4$ | $4.1 \pm 0.62$ | $3.58 \pm 1.86$ | $1.11 \pm 2.76$ | $0.29 \pm 1.74$ | $-0.23 \pm 1.32$ |
| CURL | $8.33 \pm 1.67$ | $6.01 \pm 0.28$ | $4.61 \pm 1.34$ | $1.83 \pm 2.3$ | $-0.12 \pm 0.29$ | $-1.18 \pm 0.43$ |
| GMC | $6.29 \pm 0.45$ | $5.36 \pm 1.72$ | $1.97 \pm 1.76$ | $0.75 \pm 2.56$ | $-1.74 \pm 2.96$ | $-1.08 \pm 3.09$ |
| AMDF | $2.6 \pm 1.88$ | $2.74 \pm 1.04$ | $1.98 \pm 1.55$ | $-1.35 \pm 1.55$ | $-1.32 \pm 1.97$ | $-2.3 \pm 2.12$ |
| CORAL | $5.22 \pm 2.71$ | $4.64 \pm 2.74$ | $3.3 \pm 2.01$ | $0.77 \pm 1.75$ | $-0.14 \pm 0.24$ | $0.11 \pm 0.52$ |
| MetricMM | $\mathbf{9.2 \pm 0.94}$ | $\mathbf{7.18 \pm 2.01}$ | $\mathbf{6.79 \pm 1.56}$ | $\mathbf{5.79 \pm 1.27}$ | $\mathbf{2.97 \pm 2.65}$ | $\mathbf{3.23 \pm 2.2}$ |

