# OpenReview forum: "Geometry of Uncertainty: Learning Metric Spaces for Multimodal State Estimation in RL"
_ICLR.cc/2026/Conference — ICLR 2026 Poster_

### Official Review · Reviewer_si1i · 2025-10-28

**Soundness:** 3
**Presentation:** 3
**Contribution:** 3
**Rating:** 6
**Confidence:** 3

**Summary:**

Authors propose a new latent metric (MetricMM) for sensor fusion and state estimation. The key idea is that uncertainty can be inferred from distance thereby ridding the need for explicit uncertainty. Empirically this idea proved robust to different classes of noise in a dozen of RL control domains that other methods are brittle to.

**Strengths:**

Originality and significance: The idea of utilizing the distance in latent metric space as a measure of sensor consistency thus reliability is novel as far as I am aware of. I like the motivation: it is new and simple (in a positive sense); it bypasses the noise modeling in sensor fusion which is indeed a pain unless one assumes Gaussian.

Quality and clarity: Experimental setup are thorough and results are well organized and presented.

**Weaknesses:**

I lean towards accept because no detrimental weaknesses stand out to me, though I have a few questions below that will help me become more confident.

**Questions:**

1. If I understand correctly, the noise is applied per frame by chance. What if the noise is temporally dependent, e.g., in the case of the "failure" noise, it is likely that a camera artifact persist for a few consecutive frames, "sticky artifacts" per se. How does that impact your methods and baselines?
2. How are the seven families of perturbation designed? Puzzle and texture seems too artificial to be application-relevant.
3. Each loss in eq (9) makes sense intuitively, though it would be more informative to show their role through a small ablation study even just in one domain and a few noise levels.

Decision irrelevant suggestions:
1. Table titles are cryptic. I understand you have lots of results to show (which is good signs), but it would be really helpful to expand in the title on the takeaway in natural language, at least those tables in the main body.

---

> ### Author Response · Authors · 2025-11-20
>
> We thank the Reviewer for the feedback and the suggested improvements. Below some comments on the requested changes.
>
> **Persistent noise:** We thank the reviewer for the suggestion. We have included a persistent sensor failure experiment in the updated manuscript. The results confirm the main message of the paper. When one sensor is persistently perturbed, its observation embeddings will consistently be mapped away from the predicted states and thus consistently discarded. Note that, in the persistent noise scenario, the total number of perturbed time steps increases and thus overall performances are comparably lower than the purely independent case.
>
> **Corruptions references:** Our goal in designing the corruption families was to evaluate robustness of the proposed method independently of the exact functional form of the noise. We therefore constructed a set of perturbations that span several qualitatively different failure modes. These are standard statistical perturbations that test typical (Gaussian) and extreme, sparse, dimension-wise disturbances (Salt-and-Pepper / “Impulse” noise). They are widely used in robustness benchmarks for vision models, for example in [1]. Patches and Texture are used to evaluate robustness to partial occlusion (Patches) and background/appearance shifts (Texture). Similar ideas have been explored in RL and control from pixels [2, 3] and works on generalization and augmentation in visual RL [4, 5], sometimes under names such as occlusion, background shift, or concept drift. Sensor Failure represents, arguably, the most extreme perturbation scenario in which one of the sensors fails completely, examples of its usage include [6, 7]. In Hallucination noise, the observation remains “plausible” and in-distribution at the single-modality level but is inconsistent with the true underlying state, so it must be rejected by comparing across modalities. This is inspired by adversarial robustness work in RL where corrupted observations are visually realistic but misleading [8]. Lastly, the Puzzle noise is indeed a quite artificial form of perturbation. Examples of papers dealing with it are: [9, 10]. It, however, offers an interesting form of perturbation in the particular case of convolutional networks. local patches remain intact and thus activate the same filters, but in a permuted spatial arrangement. Robustness in this setting requires the model to detect and downweight permutations of otherwise valid local features. We have added these references to the updated document.
>
> *[1]: Hendrycks, Dan, and Thomas Dietterich. "Benchmarking neural network robustness to common corruptions and perturbations." arXiv preprint arXiv:1903.12261 (2019).*
>
> *[2]: Philipp Becker, Sebastian Mossburger, Fabian Otto, and Gerhard Neumann. Combining re-
> construction and contrastive methods for multimodal representations in rl. arXiv preprint
> arXiv:2302.05342, 2023.*
>
> *[3]: Xiao Liu, Yifan Zhou, Shuhei Ikemoto, and Heni Ben Amor. α-mdf: An attention-based multimodal
> differentiable filter for robot state estimation. In 7th Annual Conference on Robot Learning, 2023.*
>
> *[4]: Grigsby, Jake, and Yanjun Qi. "Measuring visual generalization in continuous control from pixels." arXiv preprint arXiv:2010.06740 (2020).*
>
> *[5]: Hansen, Nicklas, and Xiaolong Wang. "Generalization in reinforcement learning by soft data augmentation." 2021 IEEE International Conference on Robotics and Automation (ICRA). IEEE, 2021.*
>
> *[6]: Petra Poklukar, Miguel Vasco, Hang Yin, Francisco S Melo, Ana Paiva, and Danica Kragic. Geo-
> metric multimodal contrastive representation learning. In International Conference on Machine
> Learning, pp. 17782–17800. PMLR, 2022.*
>
> *[7]: Skand, Skand, et al. "Simple Masked Training Strategies Yield Control Policies That Are Robust to Sensor Failure." 8th Annual Conference on Robot Learning. 2024.*
>
> *[8]: Zhang, Huan, et al. "Robust deep reinforcement learning against adversarial perturbations on state observations." Advances in neural information processing systems 33 (2020): 21024-21037.*
>
> *[9]: Bucci, Silvia, et al. "Self-supervised learning across domains." IEEE Transactions on Pattern Analysis and Machine Intelligence 44.9 (2021): 5516-5528.*
>
> *[10]: Noroozi, Mehdi, and Paolo Favaro. "Unsupervised learning of visual representations by solving jigsaw puzzles." European conference on computer vision. Cham: Springer International Publishing, 2016.*

---

> > ### Author Response · Authors · 2025-11-20
> >
> > **Ablation:** We thank the reviewer for this valuable suggestion. We have added an ablation study in Section 5.1 of the revised manuscript using a simple one-dimensional pendulum environment to isolate the contribution of the different loss components. Specifically, we evaluate the effect of removing the invariance term ($L_{inv}$) and the metric learning terms ($L_+$ and $L_-$). The new Figure 6 reports the resulting performance under increasing levels of sensor failure perturbations. As shown, in noise-free conditions all variants achieve comparable performance, but performance deteriorates rapidly as perturbations increase. While SAC can learn a policy even when the representations are not temporally aligned or invariant across modalities, the aggregation mechanism in Equation (3) becomes unreliable once either of these components is removed. This confirms that both the invariance and metric terms are essential for maintaining robustness under sensor corruptions.
> >
> > **Cryptic table titles:** We thank the reviewer for this suggestion. We have revised the tables to use more descriptive titles that explicitly state the key takeaway in natural language.

---

> > ### Comment · Reviewer_si1i · 2025-11-26
> >
> > Thank you for the experiments and the detailed responses. They are really helpful. I have increased my confidence score.

---

### Official Review · Reviewer_utqk · 2025-10-30

**Soundness:** 2
**Presentation:** 2
**Contribution:** 2
**Rating:** 4
**Confidence:** 4

**Summary:**

The paper proposes a novel method for learning distance metrics between states in reinforcement learning. The approach is to learn mappings from observation modalities to latent states, and measure distance using a well-define metric in the latent state space. The learned state representation can then be used as the basis for reinforcement learning algorithms. The method is tested in a series of benchmarks.

**Strengths:**

State representation learning is an important problem in reinforcement learning, and in this sense the paper makes a timely contribution. It also looks as if the proposed approach performs well in practice compared to other algorithms.

**Weaknesses:**

I believe that several concepts are not clearly explained, which makes it difficult to accurately evaluate the contribution. Mainly for this reason my opinion regarding acceptance at ICLR is on the negative side.

Apart from the cited paper by Wang et al., there are other works that explicitly learn a distance estimate between pairs of states. Concretely, the first work also measures distance as the minimum number of actions required to transition from one state to another.

State Representation Learning for Goal-Conditioned Reinforcement Learning
Steccanella & Jonsson, ECML 2022

Park, Kreiman & Levine, ICML 2024
Foundation Policies with Hilbert Representations

**Questions:**

In the definition of POMDPs, do you assume that the underlying state space S is known to the learner? (even if it is not observable)

The proposed approach is to map *each* observation modality to the same latent space, in the hope that all observation modalities derived from the same state will map to approximately the same latent state. However, the mapping from states to observation modalities is stochastic, so a deterministic map is not likely to map all modalities to the same latent state. Why do you map all observation modalities to the same latent space? An alternative would be to have a different latent space for each observation modality and aggregate the distance measures in another way.

From Equation (3) it appears as if you assume that the function \varphi_T is known, is this indeed the case? I would have expected you to *learn* an approximate transition function in the latent space. If you do not assume that \varphi_T is known, then I do not see how you can make an assumption regarding the transition error of \varphi_T, since this depends on the quality of learning.

What is the motivation for the exact form of Equation (3)?

The description of the experimental setup leaves a lot to be desired. What is the training pipeline? Do you train a distance estimate first, and then a deep RL policy, or is learning simultaneous? Is distance estimation done online or from offline data? What is the input to the deep RL algorithm (SAC), i.e. exactly how are states and state features defined?

From where did you take the seven corruption families? Can you provide a reference?

---

> ### Author Response · Authors · 2025-11-20
>
> We thank the Reviewer for the feedback and the suggested improvements. Below some comments on the requested changes.
>
> **Additional related work:** We agree with the reviewer on the existence of previous work studying metric learning applied to RL. The notion of temporal distance (or minimum action distance) has already been proposed and the two works pointed out do offer viable objectives to recover a metric space where these distances are preserved. We apologize for the confusion, we have updated the manuscript to make the comparison between our method and prior work more explicit. We would like to remark that the proposed paper builds on top of this intuition to address a very different problem from the ones tackled by previous work. In the paper, we present a new method for uncertainty estimation under unknown noise. We argue that this metric representation can offer an additional property that has been overlooked by the community so far, i.e., distance as a form of uncertainty approximation. We believe our contribution can coexist with the previous work as it offers an alternative perspective on the topic of metric learning in RL rather than being in direct competition.
>
> **Observability of the state:** We thank the reviewer for the clarification question. We do not assume that the underlying state space S is known to the learner. The POMDP tuple is used only to specify the environment theoretically. In practice, the representation module ($\varphi_{1:N}$, $\varphi_T$) is trained solely from sequences of observations and actions; the agent never accesses the true states.
>
> **Modalities invariance:** The reviewer is correct that our invariance objective (Eq. (8)) encourages sensory observations generated by the same underlying state to be mapped to (approximately) the same latent vector in Z. The core idea of our approach is to train this state estimation module on clean data (as stated at the beginning of Sec. 4.3), and then interpret relative distances in Z as a proxy for uncertainty at test time when observations may be perturbed. The reviewer is correct in pointing out that we do assume injectivity between the observations and the true unobservable state of the system. Going beyond this assumption does require a modification of the representation maps. Invariance across modalities is nevertheless crucial for our fusion mechanism in Eq. (3): distances between modality-specific embeddings are only meaningful if these embeddings live in the same latent space with a shared geometry. Without this alignment, one would need an additional, learned projection or comparison rule to relate different modality spaces before aggregation, reintroducing a complex fusion module that could itself be brittle under unseen perturbations. Our choice of a shared latent space with an invariance loss is precisely what allows the simple IDW rule to act as a robust and interpretable sensor fusion mechanism.
>
> **Learning the transition:** We apologize for the lack of clarity here. The latent transition module $\varphi_T$ is indeed learned via the optimization of Equation (7). In Equation (3), we refer to $\hat{z}$ as the embedding estimated through this learned latent transition module and with $z_i$ as the learned embedding of each modality encoding. The reviewer is right in general on the error being dependent on the quality of learning. The reviewer is correct that the prediction error necessarily depends on the quality of learning. Our transition error assumption is intended as a statement about the trained model: after training, for in-distribution transitions, we assume that $\hat{z}_t$ lies within a bounded region around the latent encodings of the actual next observations. Empirically, this is a common and reasonable behavior of one-step prediction models; errors tend to remain relatively small over short horizons and grow mainly when rolling out predictions without correction over many steps (see, e.g., Ross et al., 2011). In our setting, we explicitly avoid long open-loop rollouts: at every time step we re-anchor the state estimate by fusing $\hat{z}_t$ with the current observations via Equation (3), which prevents error from compounding over time. Note that the $\epsilon$ defined in the transition error assumption is not used in practice. It is a way to specify that in the paper we assume a transition model to have a local error.
>
> *Stephane Ross, Geoffrey Gordon, and Drew Bagnell. A reduction of imitation learning and structured prediction to no-regret online learning. In Proceedings of the fourteenth international conference on artificial intelligence and statistics, pp. 627–635. JMLR Workshop and Conference Proceedings, 2011.*

---

> > ### Author Response · Authors · 2025-11-20
> >
> > **Motivation of Equation (3):** Assuming access to a latent space where Euclidean distances correlate with temporal proximity allows us to interpret distance as a proxy for uncertainty.  The prediction error assumption effectively tells us that the true embedding is within a small region of the predicted one. When new observations become available, if they have been severely corrupted, they will result in random embeddings potentially very far away from the prediction. On the contrary, uncorrupted observations will lie around the true embedding and thus considerably closer to the transition prediction. Scaling these embeddings by their relative inverse distance results in a very low weight on the far-away embeddings (the probably corrupted ones). This aggregation is akin to a maximum a posteriori estimation by assuming the probability of the true state conditioned on each observation to be modeled by an isotropic Gaussian (maximum-entropy) with a covariance proportional to their relative distance from the predicted embedding. This algebra is only possible on a space where distance is correlated with temporal dependency thus requiring the additional optimization of Objectives (5) and (6) and when the observations' embeddings live in the same space (Objective (8)).
> >
> > **Training pipeline:** We apologize for the confusion. The proposed representation is trained end-to-end with a Soft Actor Critic (SAC) agent from the environment's observations in an online fashion. Nevertheless, pretraining the representation on an offline dataset would be possible as well in practice. During training, we make use of the original observations without any noise. As such, the input to the SAC agent is computed by means of Equation (4). After training, we test the performance of the proposed model against the baselines described in Section 5 on 50 trajectories subject to sensor perturbation with varying levels of probability. In this case, the representation (the input to SAC) is computed by means of Equation (3). Note that, in the absence of noise and with a fully trained model, Equations (3) and (4) become equivalent. The nature of the observations depends on the environment. In the Mujoco suite, we exploit RGB images and depth images as modalities. In the Fetch suite, we additionally include point clouds. In the updated manuscript, we have included an additional experiment to ablate different components of the loss. There, we make use of images and sounds generated by the pendulum swinging. More details on the experimental setup are described in Appendix A.1.

---

> > > ### Author Response · Authors · 2025-11-20
> > >
> > > **Corruption references:** Our goal in designing the corruption families was to evaluate robustness of the proposed method independently of the exact functional form of the noise. We therefore constructed a set of perturbations that span several qualitatively different failure modes. These are standard statistical perturbations that test typical (Gaussian) and extreme, sparse, dimension-wise disturbances (Salt-and-Pepper / “Impulse” noise). They are widely used in robustness benchmarks for vision models, for example in [1]. Patches and Texture are used to evaluate robustness to partial occlusion (Patches) and background/appearance shifts (Texture). Similar ideas have been explored in RL and control from pixels [2, 3] and works on generalization and augmentation in visual RL [4, 5], sometimes under names such as occlusion, background shift, or concept drift. Sensor Failure represents, arguably, the most extreme perturbation scenario in which one of the sensors fails completely, examples of its usage include [6, 7]. In Hallucination noise, the observation remains “plausible” and in-distribution at the single-modality level but is inconsistent with the true underlying state, so it must be rejected by comparing across modalities. This is inspired by adversarial robustness work in RL where corrupted observations are visually realistic but misleading [8]. Lastly, the Puzzle noise is indeed a quite artificial form of perturbation. Examples of papers dealing with it are: [9, 10]. It, however, offers an interesting form of perturbation in the particular case of convolutional networks. local patches remain intact and thus activate the same filters, but in a permuted spatial arrangement. Robustness in this setting requires the model to detect and downweight permutations of otherwise valid local features. We have added these references to the updated document.
> > >
> > > *[1]: Hendrycks, Dan, and Thomas Dietterich. "Benchmarking neural network robustness to common corruptions and perturbations." arXiv preprint arXiv:1903.12261 (2019).*
> > >
> > > *[2]: Philipp Becker, Sebastian Mossburger, Fabian Otto, and Gerhard Neumann. Combining re-
> > > construction and contrastive methods for multimodal representations in rl. arXiv preprint
> > > arXiv:2302.05342, 2023.*
> > >
> > > *[3]: Xiao Liu, Yifan Zhou, Shuhei Ikemoto, and Heni Ben Amor. α-mdf: An attention-based multimodal
> > > differentiable filter for robot state estimation. In 7th Annual Conference on Robot Learning, 2023.*
> > >
> > > *[4]: Grigsby, Jake, and Yanjun Qi. "Measuring visual generalization in continuous control from pixels." arXiv preprint arXiv:2010.06740 (2020).*
> > >
> > > *[5]: Hansen, Nicklas, and Xiaolong Wang. "Generalization in reinforcement learning by soft data augmentation." 2021 IEEE International Conference on Robotics and Automation (ICRA). IEEE, 2021.*
> > >
> > > *[6]: Petra Poklukar, Miguel Vasco, Hang Yin, Francisco S Melo, Ana Paiva, and Danica Kragic. Geo-
> > > metric multimodal contrastive representation learning. In International Conference on Machine
> > > Learning, pp. 17782–17800. PMLR, 2022.*
> > >
> > > *[7]: Skand, Skand, et al. "Simple Masked Training Strategies Yield Control Policies That Are Robust to Sensor Failure." 8th Annual Conference on Robot Learning. 2024.*
> > >
> > > *[8]: Zhang, Huan, et al. "Robust deep reinforcement learning against adversarial perturbations on state observations." Advances in neural information processing systems 33 (2020): 21024-21037.*
> > >
> > > *[9]: Bucci, Silvia, et al. "Self-supervised learning across domains." IEEE Transactions on Pattern Analysis and Machine Intelligence 44.9 (2021): 5516-5528.*
> > >
> > > *[10]: Noroozi, Mehdi, and Paolo Favaro. "Unsupervised learning of visual representations by solving jigsaw puzzles." European conference on computer vision. Cham: Springer International Publishing, 2016.*

---

### Official Review · Reviewer_Hf6h · 2025-10-31

**Soundness:** 3
**Presentation:** 3
**Contribution:** 3
**Rating:** 6
**Confidence:** 3

**Summary:**

The paper addresses the important problem of robust state estimation from multimodal, noisy observations in RL. It proposes METRICMM, a method that learns a shared latent space and uses a simple geometric fusion rule, Inverse Distance Weighting (IDW), to combine sensor estimates. The core contribution is to reframe the problem of sensor uncertainty from a probabilistic one to a geometric one. The experimental results demonstrate a clear improvement in robustness over a wide range of baselines.

**Strengths:**

The primary strength of this work is its core conceptual shift. Instead of relying on Bayesian filtering, which requires restrictive priors or generative models , METRICMM recasts uncertainty in geometric terms.

The inverse distance weighting (IDW) fusion mechanism is a direct and elegant consequence of the geometric formulation. The dynamics prediction z^t​ acts as a reliable "anchor." Any sensor encoding that is geometrically distant from this anchor is naturally identified as noise and its contribution is automatically suppressed. This is far simpler and, as the evidence suggests, more robust than learned fusion mechanisms like attention ( \alpha-MDF baseline ) or simple concatenation/linear combinations.

METRICMM consistently shows a much "flatter" degradation slope, maintaining high performance even as corruption frequency increases, while all baselines collapse quickly. Empirical results, albeit of simple benchmarks, are quite strong.

**Weaknesses:**

1) Naive Temporal Distance Loss: The paper's stated goal is to learn a space where distances correlate with the minimum number of actions required to transition between them. However, the actual loss function used is a massive oversimplification. This loss function does not model the minimum number of actions it models all single-step transitions are equidistant. It forces the distance between any two consecutive states to be 1, regardless of the optimality action taken. This is a naive implementation that contradicts the paper's core motivation.

2) I think the above formulation will be optimistic when transitions are stochastic.

3) The paper's core claim is that its learned metric representation is superior. However, the experiments confound this with a hard-coded, non-learned fusion rule (IDW). The high performance could also be due to the simple outlier-rejection property of IDW and not the learned representation. What will happen if the authors take the learned encoders from the strongest baselines (\alpha-MDF or CORAL) and, at test time, replace their learned fusion module with the paper's exact IDW fusion rule?

**Questions:**

See weaknesses

---

> ### Author Response · Authors · 2025-11-20
>
> We thank the Reviewer for the feedback and the suggested improvements. Below some comments on the requested changes.
>
> **Naive Temporal Distance Loss:** We thank the reviewer for raising this concern. Our goal is not to exactly regress the minimum number of actions between any pair of states, but to learn a latent space in which distances are monotonically related to temporal proximity. The temporal metric objective we use (a linear combination of Eqs. (5) and (6)) is a standard surrogate for this, and closely follows prior work on metric learning for RL representations. In particular, Equation (8) in Park et al. (2023) and Equation (6) in Park et al. (2024) define a hinge-based variant of the same idea, while Equation (12) in Wang et al. (2023) introduces a Lagrangian formulation for a quasimetric version of this loss. Consistent with these works, we do not claim novelty in the metric-learning objective itself; rather, we build on this line of work and apply it to a complementary problem: robust state estimation under noisy multimodal observations. Intuitively, the loss has two components: (i) it enforces that adjacent states (one-step transitions) lie at approximately unit distance, and (ii) it pushes non-adjacent states apart. In a normed latent space, this combination, together with the triangle inequality, leads to embeddings where the distance between two states reflects how many local steps are needed to connect them, i.e., an approximation of the minimum-action distance on the underlying transition graph. This surrogate is exactly the mechanism used in Wang et al. (2023) to approximate shortest-path distances in goal-reaching tasks, and we refer the reviewer there for a more detailed analysis. Note that this representation aims to reflect temporal distances in the transition graph, independently of the optimality of the policy. Our contributions lie in how this metric is then exploited for uncertainty-aware multimodal fusion, not in claiming a new or exact formulation of minimum-action distance.
>
> *Seohong Park, Oleh Rybkin, and Sergey Levine. Metra: Scalable unsupervised rl with metric-aware abstraction. arXiv preprint arXiv:2310.08887, 2023.*
>
> *Seohong Park, Tobias Kreiman, and Sergey Levine. Foundation policies with hilbert representations. arXiv preprint arXiv:2402.15567, 2024.*
>
> *Tongzhou Wang, Antonio Torralba, Phillip Isola, and Amy Zhang. Optimal goal-reaching reinforcement learning via quasimetric learning. In International Conference on Machine Learning, pp. 36411–36430. PMLR, 2023.*
>
> **Stochastic transitions:** We agree with the reviewer’s concern about stochastic transitions. MetricMM does assume determinism in this regard. We have updated the manuscript to include a limitation section (5.2) discussing this shortcoming. Additionally, we have included a stochastic one-dimensional pendulum experiment to highlight this. We empirically show that moderate stochasticity can still be handled by the proposed model and that MetricMM retains its noise robustness (compared to a ConCat baseline). The metric representation, in fact, learns to embed the next-state distribution at a unit distance and the transition converges to the barycenter of such a distribution. In practice, for high levels of noise and high-dimensional spaces, this might become unfeasible and would require an extension of the method. This might include a modification of the notion of distance to an appropriate divergence and a stochastic transition model. We believe this could be interesting to explore in the future.

---

> > ### Author Response · Authors · 2025-11-20
> >
> > **Baselines with IDW:** We understand the reviewer’s skepticism regarding the proposed fusion module and the concern that the gains might stem mainly from the inverse-distance weighting (IDW) rule rather than from the learned representation. Our goal, however, is precisely to show that the simple IDW rule only works because of the specific geometry enforced by MetricMM, not in isolation. The possibility of performing IDW over sensor embeddings fundamentally relies on two properties of the latent space Z: (i) cross-modal invariance, so that different sensors produce comparable embeddings for the same underlying state, and (ii) metric structure aligned with temporal distance, so that distances in Z reflect meaningful state similarity rather than arbitrary feature similarity. Without both properties, distances are not a valid measure of reliability/uncertainty. For example, if a trajectory is embedded as a tightly wound spiral in R^2, it is easy to obtain points that are geometrically closer to the “next turn” of the spiral than to nearby points on the same arc; in such a space, Euclidean distance does not reflect temporal proximity and IDW would be misleading. Architectures like α-MDF and CORAL train their encoders jointly with their own fusion mechanisms and are not explicitly constrained to be metric-aware in this sense. Simply replacing their learned fusion block with IDW would therefore apply IDW in a space that was never trained to support this geometric interpretation of uncertainty. Below is a table reporting the average return of α-MDF and CORAL on a Fetch Slide environment with increasing levels of Gaussian perturbation both in their original form (ORG) and with a replaced IDW aggregation (IDW). Because the attention module of α-MDF resembles a soft version of IDW, its performance in the case of low noise is not completely random yet it quickly degrades with added noise (distances are meaningless non-locally). CORAL, on the other hand, cannot handle the change in aggregation.
> >
> > A-MDF, FetchSlide, Gaussian:
> > | Noise | 0.1 | 0.25 | 0.5 | 0.75 | 0.9 | 0.99 |
> > | --- | --- | --- | --- | --- | --- | --- |
> > | IDW | -0.89  | -1.30 | -1.21 | -1.88 | -1.00 | -1.92 |
> > | ORG |  4.54 | 3.96 | 4.02 | 1.47 | 0.99 | 1.66 |
> >
> > CORAL, FetchSlide, Gaussian:
> > | Noise | 0.1 | 0.25 | 0.5 | 0.75 | 0.9 | 0.99 |
> > | --- | --- | --- | --- | --- | --- | --- |
> > | IDW | -3.33 | -3.12 | -3.02 | -3.30 | -3.68 | -3.77 |
> > | ORG | 3.37  | 3.09  | 1.02  | 0.29  | -1.01 | -0.68 |

---

### Official Review · Reviewer_PMUL · 2025-11-04

**Soundness:** 3
**Presentation:** 3
**Contribution:** 3
**Rating:** 6
**Confidence:** 4

**Summary:**

This paper proposes a novel method for robust multimodal state estimation in POMDPs by learning a structured latent representation. The core idea is to create a metric space where the Euclidean distance between latent states directly correlates with the minimum number of actions required to transition between them.

All sensor modalities are encoded into this shared space, providing a geometric interpretation of uncertainty that avoids the need for explicit probabilistic noise models. The system fuses information by weighting each modality's contribution based on its inverse distance to the predicted state, thereby down-weighting corrupted or unreliable sensor data. Empirical results demonstrate that this approach significantly enhances an RL agent's performance and robustness against unseen sensor corruptions.

**Strengths:**

- The paper introduces a novel geometrical view for handling state representation and uncertainty in POMDPs. By refraining uncertainty in geometric terms it bypasses the complexities and assumptions of traditional probabilistic models. This idea could inspire a new direction of research for handling partial observability and uncertainty in RL.

- The paper is well written and the methodology is sound. Including implementation details that could make future reproduction of the algorithm and empirical results easy.

- The experimental results are extensive and provide convincing evidence of the method's effectiveness.

**Weaknesses:**

- The paper fail in citing previous work from Steccanella et al. (2022), "State Representation Learning for Goal-Conditioned Reinforcement Learning". That work appears to be the first to propose the idea of minimum number of actions distance and motivates very similar objectives for learning an embedding space where distance between states in this embedding space approximates the minimum action distance, by means of leveraging local constraints and the useful upper-bound of the trajectory distance instead of simply trying to maximize the log distance. Please discuss and include this citation in your work.

- The paper should make more clear to the reader that this approach assume deterministic dynamics. Is not clear to me how this approach will behave in the case of stochastic dynamics where the distance between states becomes an expectation and the minimum action distance will provide just a lower bound on that. And a broader discussion of this limitation is needed. The authors should explicitly state this assumption and dedicate a discussion to this limitation, outlining how the framework will behave in this scenario.

- The paper uses a symmetric metric (Euclidean distance) to approximate the Minimum Action Distance, which is inherently asymmetric in most realistic environments. For instance, in environments with irreversible or asymmetric dynamics. By enforcing a symmetric metric, the model is forced to learn an inaccurate representation that cannot capture such crucial, directional aspects of the environment's dynamics. As noted in prior work (Steccanella et al., 2022), this forces only to learn a symmetric approximation of the true Minimum Action Distance. A more thorough discussion is needed to acknowledge this trade-off between simplicity and representational fidelity.

**Questions:**

See Weaknesses.

---

> ### Author Response · Authors · 2025-11-20
>
> We thank the Reviewer for the feedback and the suggested improvements. Below some comments on the requested changes.
>
> **Citing previous work:** We thank the reviewer for pointing out relevant related work. We have updated the manuscript to include the citation. We agree that metric learning in the context of RL has been previously studied and it is still an active research topic. Most of the works (including Steccanella et al. (2022)) have focused on these representations specifically to improve policies (reactive or planning) or value estimates. The primary goal of this paper is to extend this formulation to the adjacent field of state estimation under unknown perturbations. We believe this provides a complementary, non-probabilistic perspective on robustness in partially observable RL.
>
> **Deterministic dynamics assumption:** We thank the reviewer for the suggestion. We have updated the manuscript with a limitation section discussing the dynamics assumption. We agree that fully stochastic dynamics pose challenges both for the underlying metric (minimum-action distance becomes an expectation) and for learning a deterministic transition model. Moderately stochastic dynamics can still be handled by MetricMM in practice. The latent transition model learns to predict the barycenter of the next-state distribution, while the embeddings of actual next states form a cluster at approximately unit distance around this anchor. We added an additional experiment on a stochastic pendulum swing-up task (Sec. 5.2), where MetricMM is compared to a concatenation baseline under varying process noise. As stochasticity increases, overall performance degrades for all methods, but MetricMM retains a robustness advantage and continues to provide useful uncertainty estimates for state fusion.
> We also note that strongly stochastic environments would likely require a higher-dimensional latent space to faithfully represent multiple possible futures, and a deterministic transition model will eventually become inaccurate in this regime. Extending the framework to explicitly probabilistic transitions (e.g., replacing the norm with a divergence between predictive distributions) is an interesting direction for future work.
>
> **Discussion on symmetry:** We have expanded the discussion in Sec. 5.2 to address this point. As highlighted in Steccanella et al. (2022), Park et al. (2023), and Wang et al. (2023), representing temporal distances of a general MDP in a normed (symmetric) latent space is inherently approximate, since minimum-action distances are typically asymmetric. Our formulation shares this limitation: MetricMM learns a symmetric approximation of a fundamentally quasimetric structure. The main reason for this design choice is optimization simplicity and stability: using a norm-based metric yields a straightforward geometric loss that works well with standard deep RL training. MetricMM then uses this metric structure primarily to derive an uncertainty-like quantity for multimodal fusion. In asymmetric environments, this uncertainty will inherit some approximation error from the symmetric metric; an explicitly asymmetric distance could, in principle, provide a tighter and more faithful estimate, and we see this as a promising extension. Importantly, this asymmetry is not catastrophic in our setting. The SAC policy consumes the latent representation through an expressive neural network, and is not forced to obey any symmetry constraints itself. Empirically, all of our tasks are at least partially asymmetric (e.g., pushing the cube in the Fetch environments), yet MetricMM still yields strong performance and robustness. This suggests that a symmetric latent metric can be sufficiently informative for both control and robust fusion even in non-symmetric MDPs.

---

### Author Response · Authors · 2025-11-20
**Updated Document**

We thank all reviewers for their detailed and constructive feedback. A revised version of the paper has been uploaded, with changes marked in blue. Key updates include expanded related work, a new ablation study on the loss components, an additional experiment on a stochastic pendulum environment, and a dedicated limitations section discussing symmetry and stochasticity.

---

### Author Response · Authors · 2025-12-02
**Rebuttal Summary**

We once again thank the reviewers for their feedback. Due to the recent events, there was limited opportunity to engage in a thorough discussion of the paper’s content. Below we briefly summarize the main concerns raised and how they are addressed in the current revision of the manuscript (already uploaded, with changes marked in blue).

**Related work:**
Several reviewers pointed out missing related work on learning minimum-action distances. We have added these works explicitly in the related work section. We now clarify that we do not claim novelty in the metric-learning objective itself; instead, we build on this line of work and apply it to a different problem: robust multimodal state estimation and sensor fusion under unknown noise, where distance is used as a proxy for uncertainty. We also make clear that our temporal distance loss is a standard surrogate (monotonic in temporal proximity) rather than an exact regression of the minimum number of actions.

**Deterministic dynamics & symmetry:**
We now include a limitations section explicitly stating the determinism and symmetry assumptions and discussing how MetricMM behaves when these are violated. We added a stochastic pendulum swing-up experiment to probe these conditions. As noise increases, performance degrades for all methods, but MetricMM retains a robustness advantage and continues to exhibit meaningful “distance-as-uncertainty” behavior. We also expanded the discussion on using a symmetric metric (Euclidean norm) to approximate inherently asymmetric minimum-action distances. We acknowledge that this yields a symmetric approximation of a quasimetric structure, in line with prior work. We explain why we chose this design (optimization simplicity and stability) and why it is acceptable in our setting: the SAC policy is not constrained to be symmetric, and empirically our tasks include asymmetric dynamics (e.g., pushing in Fetch), yet MetricMM still provides strong control and robustness.

**Corruption families:**
We added references connecting the noise types to the existing literature. We also added a persistent sensor failure experiment, where a modality is corrupted for multiple consecutive time steps. As expected, overall performance decreases relative to i.i.d. noise (more total corrupted steps), but MetricMM continues to downweight the persistently corrupted modality and maintains a clear robustness margin over baselines.

**Learned representation vs. IDW fusion rule:**
One reviewer asked whether the performance gains might come primarily from the inverse-distance weighting (IDW) fusion rule rather than from the learned metric representation. IDW only becomes meaningful if the latent space satisfies two properties: (1) cross-modal invariance and (2) a metric structure aligned with temporal proximity. Without these, distances are not a valid proxy for reliability/uncertainty. We ran additional experiments where we took strong multimodal baselines and replaced their learned fusion modules with our IDW rule at test time. This consistently degraded their performance, especially under stronger perturbations, supporting our claim that the benefit does not come from IDW alone but from the specific geometry enforced by MetricMM.

Overall, the revised manuscript incorporates all requested citations and clarifications, adds new experiments on stochastic dynamics, persistent noise, and ablations, and explicitly discusses the main modeling limitations (determinism and symmetric metrics). We hope this summary helps the area chair quickly see how the current version addresses the reviewers’ concerns.

---

### Meta-Review · Area_Chair_6aBy · 2026-01-07

**Summary:**

The reviewers’ main concerns relate to positioning and interpretation rather than correctness: how the work relates to prior metric-learning literature, how the temporal distance objective should be understood, and under which assumptions (determinism, symmetry, moderate stochasticity) the approach is expected to hold. There are also questions about whether the gains stem from the learned representation or mainly from the inverse-distance fusion rule, and about robustness under different corruption regimes.

The rebuttal clarifies these points, aligns the formulation with prior work, explicitly states the main assumptions and limitations, and adds targeted experiments and ablations to support the empirical claims. As a result, the remaining concerns are primarily about scope and interpretation rather than technical soundness.

On this basis, I view the outstanding issues as reasonable limitations rather than blockers, which informs my recommendation to accept.

**Reviewer Concerns:**

Several of the main concerns raised by the reviewers are addressed in the response.

Prior work and novelty (PMUL, utqk). Both reviewers point out missing references. The authors add these references, and clarify that they do not claim novelty for the metric-learning objective itself. Instead, they position their contribution as using such a metric as an uncertainty signal.

Temporal distance objective (Hf6h, utqk): the one-step/unit-distance loss may be too naive to represent true minimum-action distance. The response makes it clear that the objective is meant as a surrogate for temporal proximity, consistent with prior work, rather than an exact regression target, which removes the apparent mismatch between the stated goal and the implemented loss.

Symmetry of the metric (PMUL). The concern about approximating an inherently asymmetric distance with a symmetric norm is acknowledged. The authors treat this as a conscious design trade-off for stability and show empirically that it does not break the method on partially asymmetric tasks.

Corruptions and temporal dependence (si1i, utqk). The authors justify the corruption families with references and add a persistent sensor failure experiment, showing that the model continues to downweight a bad modality over consecutive steps.


----

Concerns that are only partially resolved

How close is the learned metric to true minimum-action distance? (Hf6h, utqk). The authors clarify that the metric is only a surrogate for temporal proximity, and they no longer claim an exact match. Still, there is no formal characterization of how well it approximates true minimum-action distances beyond empirical results.

Very strong stochasticity and partial observability (PMUL, utqk). The paper now clearly states that strongly stochastic settings would likely require a distributional transition model or a different geometry. This is a reasonable limitation, but it remains an open direction rather than something addressed in the current work.

**Reviewer Scores:**

See above

---

### Decision · Program_Chairs · 2026-01-26

Accept (Poster)